# Study on collateral sensitivity of tigecycline to colistin-resistant *Enterobacter cloacae* complex

**Kaixin Yu,**[1,2] **Jiming Wu,**[1] **Mingjing Liao,**[1] **Jianmin Wang,**[1] **Chunli Wei,**[1] **Wenzhang Long,**[1] **Xuemei Gou,**[1] **Yang Yang,**[1] **Jin Wang,**[1] **Xushan Liang,**[1] **Chunjiang Li,**[2,3] **Xiaoli Zhang**[1]

**ABSTRACT** The past decade has witnessed the emergence and spread of carbapenem-resistant *Enterobacter cloacae* complex (CRECC), presenting a significant clinical challenge and urgently demanding new treatment strategies against antimicrobial resistance (AMR). This study focused on the impact of tigecycline on the susceptibility of CRECC isolates to colistin and the collateral sensitivity in CRECC. Under tigecycline pressure, the resistance of five clinically isolated CRECC strains to colistin was converted from resistance to sensitivity. These mutants exhibited significantly higher expression of *acrA*, *acrB*, and *ramA* genes, with mutations in the *ramR* gene. Overexpression of *ramA* in certain mutants did not alter *ramR* expression. No mutations were identified in lipid A synthesis genes; however, *phoQ* was consistently downregulated, and *arnA* expression varied among CRECC405-resistant mutants. Low-dose colistin and tigecycline combination therapy outperformed monotherapy in antimicrobial efficacy. Overall, collateral susceptibility to tigecycline was observed in CRECC isolates with colistin resistance. The overexpression of *acrA, acrB,* and *ramA*, due to *ramR* mutations, led to tigecycline resistance. Inconsistent expression levels of lipid A synthesis genes affected lipid A modification, which in turn upregulated *arnA* expression in CRECC405-resistant mutants. Increased sensitivity to colistin was associated with the downregulation of *phoQ* and *arnA* expression.

**IMPORTANCE** Antimicrobial resistance (AMR) is escalating faster than our ability to manage bacterial infections, with antibiotic-resistant bacteria emerging as a significant public health risk. Innovative strategies are urgently needed to curb AMR dissemination. Our research identified collateral sensitivity in *Enterobacter cloacae* complex following tigecycline (TGC) resistance, resulting in newfound sensitivity to colistin (COL), a drug to which it was once resistant. Synergistic tigecycline and colistin therapy significantly suppress bacterial growth, offering a promising approach to combat infections and curb AMR progression through the strategic pairing of antibiotics with complementary sensitivities.

**KEYWORDS** CRECC, colistin, tigecycline, collateral sensitivity

Collateral sensitivity (CS) has important implications for antimicrobial research in multi-drug-resistant bacteria and opens new avenues for novel therapeutic strategies (1). The aim of this study was to investigate the collateral sensitivity of tigecycline (TGC) to colistin in clinically isolated carbapenem-resistant *Enterobacter cloacae* complex (CRECC). The in *vitro* efficacy of tigecycline and colistin alone or in combination was fully evaluated to provide new theoretical insights into the clinical treatment of CRECC.

The *Enterobacter* clinical isolates commonly encountered in practice, often referred to as "*E. cloacae*," actually constitute a heterogeneous group known as the *Enterobacter*

**Peer Reviewer** François Guerin, CHU de Rennes, Rennes, France

Address correspondence to Xiaoli Zhang, jmszxl123@163.com, or Chunjiang Li, mdjlcj07@163.com.

Kaixin Yu and Jiming Wu contributed equally to this article. The order of authors is mainly determined based on the magnitude of their contributions.

The authors declare no conflict of interest.

*cloacae* complex (ECC) (2). CRECC strains frequently harbor metal β-lactamase genes, such as NDM-1 or IMP-4 (3). The β-lactam/β-lactamase inhibitor combination ceftazidime/avibactam (CZA) has emerged as a key therapy for CRE infections. However, increasing clinical use has led to rising CZA resistance, further limiting treatment options for CRE (4). *Klebsiella pneumoniae* (*K. pneumoniae*) can develop resistance to CZA through multiple mechanisms, primarily involving mutations in KPC-2 and OmpK36 (5). Similarly, mutations in the *PiuC*, *PiuA,* and *PirA* genes of *P. aeruginosa* carrying both IMP-16 and KPC-2 carbapenemase genes can confer in *vivo* resistance to cefiderocol (6).

Tigecycline, a tetracycline derivative, and colistin, a polymyxin, have been regarded as the "last line" of treatment for infections caused by MDR gram-negative bacteria. However, the emergence of resistance has aroused widespread concern in recent years (7).

The widespread use of this cationic antimicrobial peptide has led to the emergence of colistin resistance globally (8, 9). The plasmid-mediated mobile colistin resistance gene, *mcr-9,* is highly prevalent among CRECC strains and significantly compromises the efficacy of colistin treatment (10, 11). Previous studies show that antibiotic pressure increases colistin resistance, heightening treatment failure risks (12, 13). The clinical use of colistin has been limited; hence, it is important to use colistin in combination with other antibiotics to guide clinical treatment (14, 15).

Tigecycline is a member of the glycylclines class of antibiotics, approved by the U.S. Food and Drug Administration (FDA) for the treatment of complex infections, and has good activity in the treatment of *Enterobacteriaceae* bacteria (16). The main mechanism of action of tigecycline is similar to other tetracyclines in that it acts as an inhibitor of bacterial protein translation via reversible binding to the 30S subunit of bacterial ribosomes, thus impeding bacterial growth (17). During recent years, tigecycline resistance has emerged and has mostly been identified in *Enterobacterales* (18). High expression of AcrAB-TolC and OqxAB efflux pumps plays a crucial role in tigecycline resistance (19). It was observed that the AcrAB and OqxAB efflux pumps were notably overexpressed in carbapenem-resistant tigecycline heterogeneous-resistant *E. cloacae*, in which the increased expression of their regulatory genes *ramA* and/or *soxS* was presumed to be a key factor in the heterogeneous resistance of tigecycline (20). Some studies have reported that TGC-sensitive *K. pneumoniae* develops resistance during treatment; meanwhile, increasing resistance to TGC can reduce the virulence of *K. pneumoniae* and affect its sensitivity to other antibiotics (21–23). Clinicians prefer combination therapy due to monotherapy's poor efficacy and the threat of bacterial resistance (24). Development of CS is a widespread evolutionary trade-off that occurs in many gram-negative and gram-positive bacteria (25). Antibiotic collateral sensitivity is a phenomenon in which resistance to one antibiotic leads to increased susceptibility or hypersensitivity to a different, often structurally unrelated, antimicrobial agent (26). Although the idea of CS was initially described by Bryson and Szybalski in the 1950s, who observed increased sensitivity to polymyxin B in an *Escherichia coli* strain after acquiring chloramphenicol resistance, it has now become a topic of significant interest among researchers (27). Many studies have delved into the mechanisms of CS in diverse clinically pathogenic microbes, aiming to design effective therapies and curb the resurgence of resistance (28–34). Decades of research on combination therapies have focused on physiological interactions like synergy and antagonism. However, evolutionary interactions that maintain CS across bacterial strains with diverse mutational profiles are essential for effective clinical application. Investigating the resilience of CS in established antibiotic-resistant mutants to new antimicrobials is critical, a topic that remains underexplored in clinical settings (35). Experimental studies show promising results in bacterial extinction and minimal resistance evolution. There is a significant correlation between resistance mutation patterns and drug types and sequences (36). CS-based strategies offer flexibility in combining antibiotics, including simultaneous, sequential, or cyclic administration (37).

## MATERIALS AND METHODS

### Isolation and identification of strains and antibiotic sensitivity detection

In total, 212 ECC strains were isolated from clinical specimens at a teaching hospital, among which 38 strains were CRECC. Included among these were five CRECC strains that were highly resistant to colistin but sensitive to tigecycline (CRTS): CRECC401, CRECC402, CRECC405, CRECC414, and CRECC416. These strains were used for in *vitro* tigecycline induction to verify whether CS was created (Table S1). Bacterial identification was performed using the Vitek 2 system (bioMérieux, Marcy I'Etoile, France) and MALDI-TOF mass spectrometer (Bruker, Billerica, MA, USA). The broth microdilution (BMD) method was used to determine the MIC of antibiotics. The susceptibility results for Colistin (COL) and TGC were interpreted according to the guidelines of CLSI (38) and EUCAST (39), respectively, (COL susceptible, ≤2 mg/L; COL resistant, ≥4 mg/L); (TGC susceptible, ≤0.5 mg/L; TGC resistant, >0.5 mg/L). *Escherichia coli* ATCC 25922 was used as the control strain.

### Whole genome sequencing (WGS)

Genomic DNA from five CRECC strains was extracted using the Wizard Genomic DNA Purification Kit and sequenced using Illumina HiSeq 2000 (2*150 bp paired-end) (Illumina Inc., San Diego, CA, USA) and MinION platforms (Oxford Nanopore Technologies, Oxford, UK). Genomes were assembled with Canu v.1.6 and polished with Pilon v1.22. Annotations were conducted with PGAP and Glimmer 3.02 (http://www.cbcb.umd.edu/software/glimmer/). PlasmidFinder (https://cge.food.dtu.dk/services/PlasmidFinder/), CARD, and ResFinder (https://cge.food.dtu.dk/services/resfinder/) identified plasmid sequences, resistance genes, and virulence factors, whereas ISfinder (https://www-is.bio-toul.fr/) located transposons and IS elements. BLASTn was used for sequence comparisons, and BRIG (http://brig.sourceforge.net) and Easyfig (https://github.com/mjsull/Easyfig) were used for visualization (40). The genomes are deposited in GenBank (Gene registration numbers are listed in Table S1).

### Induction experiment in *vitro*

To induce TGC resistance, the five CRECC isolates were cultured in Luria-Bertani (LB) broth at 37°C with shaking overnight, using successively doubled concentrations of TGC ranging from 0.25 to 16 mg/L. Resistant mutants from each cultivation were frozen in glycerol at –80°C before being used in the antimicrobial susceptibility testing. The most recent generation, the obtained tigecycline-resistant mutants, was designated CRECC401R, CRECC402R, CRECC405R, CRECC414R, and CRECC416R.

### Identification of gene mutations associated with colistin and TGC resistance

All strains before and after induction were tested for gene mutation by Sanger sequencing. The DNA of bacteria was extracted by the boiling method and used as the template of polymerase chain reaction to amplify the genes related to drug resistance. Gel electrophoresis validation was performed in 0.5× Tris boric acid EDTA Run Buffer (BIO-RAD) with 1.2% agarose. Primer sequences for PCR assays are listed in Table S2. The positive amplification products were sequenced using Sanger sequencing, and the sequences obtained were compared with those available on the internet using the Basic Local Alignment Search Tool (BLAST) tool (http://www.ncbi.nlm.nih.gov/blast/).

### RNA extraction and RT-qPCR

Collected bacterial cultures were used for RNA extraction of CRTS isolates and TGC-resistant mutant strains. Bacterial cultures were collected at $12,000 \times g$ for 10 min at 4°C, and the supernatant was discarded. Total RNA extraction was performed using the PureLink-TIM RNA Mini Kit (Thermo Fisher Science) following the manufacturer's protocol. The concentration and purity of RNA were assessed using ND-2000 (NanoDrop Technologies).

Reverse transcription of each RNA sample was carried out using the PrimerScript RT kit (Takara, Japan) according to the manufacturer's instructions. DEPC water was used as a blank control template. Quantitative analysis of relative gene expression levels was conducted using SYBR Green assay (MedChemExpress) on a CFX96 fluorescence quantitative polymerase chain reaction system (Bio-rad). The relative gene expression levels were calculated by the $2^{-\Delta\Delta CT}$ method, and the results were normalized to *rpoB* as a reference gene for sample comparison.

Statistical analyses were performed using GraphPad Prism 9.5 with unpaired *t*-tests and Welch's correction, and graphs were made.

## Stability test for tigecycline resistance mutants

To investigate the stability of the minimum inhibitory concentrations (MICs) of TGC-resistant mutants derived from each CRECC, these mutants were inoculated into fresh LB broth without any antibiotics and incubated at 37°C with shaking at 160 rpm. A dilution of 1:100 was performed every 12 h for a duration of 10 days (200 generations) using LB broth. Determination of drug sensitivity using the BMD after 10 days. The initial and final MICs of CRECC against TGC and COL were plotted as dumbbell bar graphs.

## Growth curves

To evaluate the growth fitness alteration of TGC-resistant mutants in comparison to their parental strains, these mutants were inoculated into fresh LB broth devoid of any antibiotics and incubated at 37°C with shaking at 160 rpm. Bacterial cultures were then inoculated into 30 mL LB medium broth with a ratio of 1:100, and the cultures were diluted in LB to reach turbidity equal to 0.5 McFarland, shaken at 37°C continuously. The absorbance of the bacterial medium at 600 nm was measured every hour. Each strain was cultured in triplicate, and the mean absorbance was calculated. Growth curves of the corresponding strains were plotted using GraphPad Prism 9.5. A *t*-test was used to determine the significance between the corresponding strains.

## Time-killing assay

To determine the bactericidal efficacy of TGC and COL, both alone and in combination, a time-killing test was conducted following established protocols (41). An overnight culture, adjusted to a McFarland standard of 0.5, was added to cation-adjusted Mueller-Hinton broth (CA-MHB) to achieve a final volume of 20 mL. The dilution ratio of bacterial cultures to broth is 1:100. The initial bacterial concentration in the conical flasks was $10^6$ CFU/mL and was incubated in a shaking incubator at 37°C, along with a growth control without added drug. Samples were collected at 0, 2, 4, 8, 12, and 24 h, serially diluted, and spread onto LB agar plates for colony formation and counting. The experiment was performed in triplicate. Results were interpreted as follows: bactericidal activity was defined as a ≥ 3 Log10 CFU/mL reduction in colony count relative to the initial inoculum; synergistic effect was defined as a ≥ 2 Log10 CFU/mL reduction in colony count at 24 h compared with the most potent single agent (41).

## RESULTS

### Characteristics of CRECC clinical isolates

From 38 CRECC strains, five isolates with high COL resistance were identified. These strains showed resistance to multiple antibiotics (Table 1). CRECC402 and CRECC416 were sensitive to Aztreonam, whereas CRECC402 and CRECC414 were sensitive to Nalidixic Acid. Additionally, CRECC402, CRECC405, CRECC414, and CRECC416 were sensitive to Norfloxacin. The MICs of COL and TGC for all five CRECC strains were determined using the broth microdilution method, revealing high resistance to COL (MIC >128 mg/L) and susceptibility to TGC. These strains exhibited homogeneous resistance to COL (Table 1), as no "skip-well" phenomenon was observed during testing. However, this method may

**TABLE 1** Antimicrobial susceptibility of five clinical isolates and five resistance-induced descendant strains

| | Antibiotics[b] | | | | | | | | | | |
|---|---|---|---|---|---|---|---|---|---|---|---|
| Isolates | TGC | COL | MEM | DOR | TIM | PIP | CTX | CZX | ATM | NA | NOR |
| Breakpoints(S-R) MIC(mg/L)[a] | | | | | | | | | | | |
| CRECC401 | 0.5 | ≥128[R] | ≥16[R] | ≥8[R] | ≥128[R] | ≥128[R] | ≥64[R] | ≥64[R] | 16[R] | ≥32[R] | ≥16[R] |
| CRECC401R | ≥8 | 1[S] | ≥16[R] | ≥8[R] | ≥128[R] | ≥128[R] | ≥64[R] | ≥64[R] | 4[S] | ≥32[R] | ≥16[R] |
| CRECC402 | 0.5 | ≥128[R] | ≥16[R] | ≥8[R] | ≥128[R] | ≥128[R] | ≥64[R] | 16[R] | ≤1[S] | ≤2[S] | ≤0.5[S] |
| CRECC402R | ≥8 | 2[S] | ≤0.125[S] | ≤0.125[S] | ≤8[S] | 8[S] | ≤1[S] | ≤1[S] | ≤1[S] | ≥32[R] | 2[S] |
| CRECC405 | 0.5 | ≥128[R] | ≥16[R] | ≥8[R] | ≥128[R] | ≥128[R] | ≥64[R] | 32[R] | ≥64[R] | ≥32[R] | 1[S] |
| CRECC405R | ≥8 | 1[S] | ≥16[R] | ≥8[R] | ≥128[R] | ≥128[R] | ≥64[R] | 8[R] | ≥64[R] | ≥32[R] | 8[I] |
| CRECC414 | 0.5 | ≥128[R] | ≥16[R] | ≥8[R] | ≥128[R] | ≥128[R] | ≥64[R] | ≥64[R] | ≥64[R] | 16[S] | 2[S] |
| CRECC414R | ≥8 | 1[S] | ≥16[R] | 4[R] | ≥128[R] | ≥128[R] | ≥64[R] | ≥64[R] | ≥64[R] | ≥32[R] | 2[S] |
| CRECC416 | 0.25 | ≥128[R] | ≥16[R] | ≥8[R] | ≥128[R] | ≥128[R] | ≥64[R] | 32[R] | ≤1[S] | ≥32[R] | 2[S] |
| CRECC416R | ≥8 | 1[S] | ≥16[R] | ≥8[R] | ≥128[R] | ≥128[R] | ≥64[R] | ≥64[R] | ≤1[S] | ≥32[R] | 16[R] |

[a]S-R represents the susceptible (S) breakpoint to the resistant (R).
[b]TGC, Tigecycline; COL, Colistin; MEM, Meropenem; DOR, Doripenem; TIM, Ticarcillin/Clavulanic Acid; PIP, Piperacillin; CTX, Cefotaxime; CZX, Ceftizoxime; ATM, Aztreonam; NA, Nalidixic Acid; NOR, Norfloxacin; S, Susceptible; R, Resistant.

not detect extremely low-frequency resistant subpopulations, potentially overlooking heteroresistance (42).

Whole genome sequencing of the five CRTS strains was performed to analyze drug resistance genes. The clinical isolates showed high homology (Fig. 1). CRECC401 and CRECC414 were identified as *Enterobacter kobei*, CRECC402 and CRECC416 as *Enterobacter asburiae*, and CRECC405 as *E. cloacae*. MLST analysis revealed that CRECC402 and CRECC405 were ST25, whereas CRECC401, 414, and 416 were ST591, ST365, and ST484, respectively (Fig. 1). These are not the epidemic sequence types ST114, ST93, ST90, and ST78 associated with ECC infections (43).

Among these strains, CRECC401, CRECC414, and CRECC416 harbored the metallo-beta-lactamase gene *bla*$_{NDM-1}$, whereas all except CRECC405 carried *bla*$_{IMP-4}$. CRECC402, CRECC414, and CRECC416 also carried *bla*$_{TEM-1}$. The two-component system genes *phoP* and *phoQ*, located on chromosomes, and the plasmid-borne *mcr-9* gene were present in CRECC405, CRECC414, and CRECC416. In our previous study, the *mcr-9* gene in CRECC402 and CRECC405 strains was located on the IncHI2/2A plasmid, with a genetic context of IS*903B*-*mcr-9*-*wbuC*-IS*26*. In CRECC414, the *mcr-9* gene was found on the pECL414-1 plasmid, flanked by IS*1B*-*mcr-9.2*-*wbuC*-IS*26* (11). In CRECC416, the *mcr-9.1* gene was located upstream of IS*903B* and downstream of IS*481* and IS*26* (44). These insertion sequences may facilitate the stable expression and dissemination of the *mcr-9* gene, potentially serving as key factors in the spread of COL resistance among CRECC strains in hospital settings. Additionally, the high-level COL resistance observed in CRECC401 and CRECC402 clinical isolates may be attributed to other mechanisms, such as the overexpression of efflux pumps (e.g., AcrAB-TolC) and biofilm formation. *E. cloacae* can

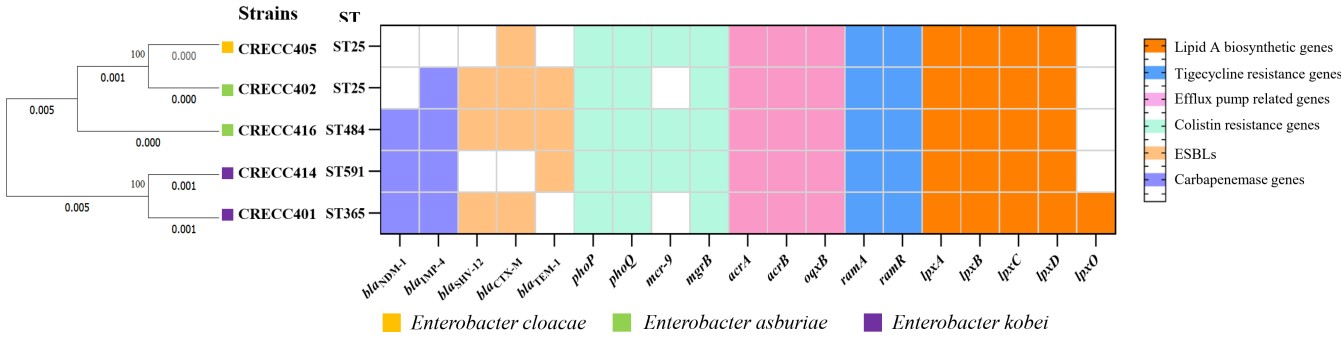

**FIG 1** Clustering analysis and resistance determinants of 5 CRECC strains from our institution. The resulting structure reflected the similarity between the sequences, and different resistance determinants present (in different colors) in each strain are shown on the right.

expel COL from cells via efflux pump systems, reducing its intracellular accumulation and conferring resistance (45). Overexpression of regulatory genes such as *soxS* and *ramA* can further enhance resistance by activating efflux pump systems (46).

Multidrug efflux pumps and their regulators are crucial in multidrug resistance. The AcrAB efflux pump, encoded by *acrA* and *acrB* genes, and the OqxAB efflux pump, key gene *oqxB*, were found in all five strains. Except for *lpxO*, observed only in CRECC401, enzymes involved in lipid A biosynthesis, including *lpxA, lpxB, lpxC, lpxD,* and *lpxL*, were conserved across all strains.

## Changes in drug resistance after tigecycline exposure

After continuous in *vitro* induction, we observed phenotypic changes in the 5 TGC-resistant mutant strains (Fig. 2A). The MIC of various drugs post-induction was determined (Table 1; Fig. 2B). Among the induced strains, a significant decrease in COL MIC (≥64-fold reduction) was common, shifting from high resistance to sensitivity. The reduction in COL MIC values may be due to the loss of *mcr* gene-carrying plasmids, as evidenced by plasmid curing experiments showing a decrease from >256 mg/L to 4 mg/L (47, 48). COL may also kill TGC-resistant mutants by binding to lipid A and disrupting their outer membrane integrity (49).

Tigecycline MIC increased significantly, with values ≥ 8 mg/L. Additionally, tigecycline-resistant strains exhibited variable changes in sensitivity to different antibiotic classes, including carbapenems and third-generation cephalosporins. Notably, the MICs of meropenem and doripenem for CRECC402R decreased significantly from initial values of ≥16 mg/L and ≥8 mg/L to ≤0.125 mg/L (≥128-fold reduction) and ≤0.125 mg/L (≥64-fold reduction), respectively. Additionally, CRECC402R exhibited increased susceptibility to piperacillin, ticarcillin/clavulanic acid, and third-generation cephalosporins (e.g., cefotetan and cefoxime) (Table 1). These changes may be associated with the loss of plasmids carrying β-lactamase genes (50). The accessory sensitivity induced by TGC exposure was prominently reflected in COL, whereas changes in resistance to other antibiotic classes were inconsistent.

## Mutation and expression of drug-resistance-related genes

We sequenced COL resistance genes, TGC resistance genes, and regulator-associated efflux pump genes to identify molecular determinants for sensitivity and resistance. Mutations were detected in both parental strains and TGC-resistant mutants within COL resistance genes (*phoP, phoQ,* and *mgrB*) and efflux pump-related genes (*acrA, acrB, oqxB, marA,* and *soxS*). However, no mutations were found in any of the genes mentioned, with the exception of *ramR* or *ramA*. An amino acid substitution from tryptophan to lysine at position 37 in the *ramA* gene was identified in CRECC414R (Table 2). Additionally, premature termination codons and insertion sequences were found in the *ramR* gene of TGC-resistant strains CRECC401R and CRECC416R, respectively (Table 2). In CRECC401R, a G to A mutation at the 555th base results in a premature termination codon at amino acid 185, truncating the original 196-amino acid sequence. CRECC416R contains an IS*26* insertion sequence, with an 820 bp sequence inserted after 127 bases, flanked by a 10 bp direct repeat sequence (Table 2).

We assessed mRNA expression changes in five isolates following TGC exposure, focusing on efflux pump genes (*acrA, acrB,* and *oqxB*), COL resistance genes (*phoQ* and *arnA*), TGC resistance genes (*ramR* and *ramA*), and lipid synthesis genes (*lpxA, lpxB, lpxC, lpxD,* and *lpxL*). Expression levels of *phoQ* and *arnA* were generally low in resistant mutants, with CRECC405R showing a higher *arnA* expression compared with CRECC405 (4.6-fold). Conversely, the relative expression of *acrA, acrB,* and *oqxB* was significantly upregulated in TGC-resistant mutants, particularly *acrB*, which increased 5-fold to 17-fold over parental strains (Fig. 3). The *ramA* gene, which regulates efflux pumps AcrAB and OqxAB, showed a significant upregulation in four TGC-resistant mutants (4-fold to 186-fold) but was downregulated in CRECC414R (0.1-fold), correlating with a missense mutation in *ramA* (Fig. 3). RamR expression decreased in CRECC401R, CRECC414R, and

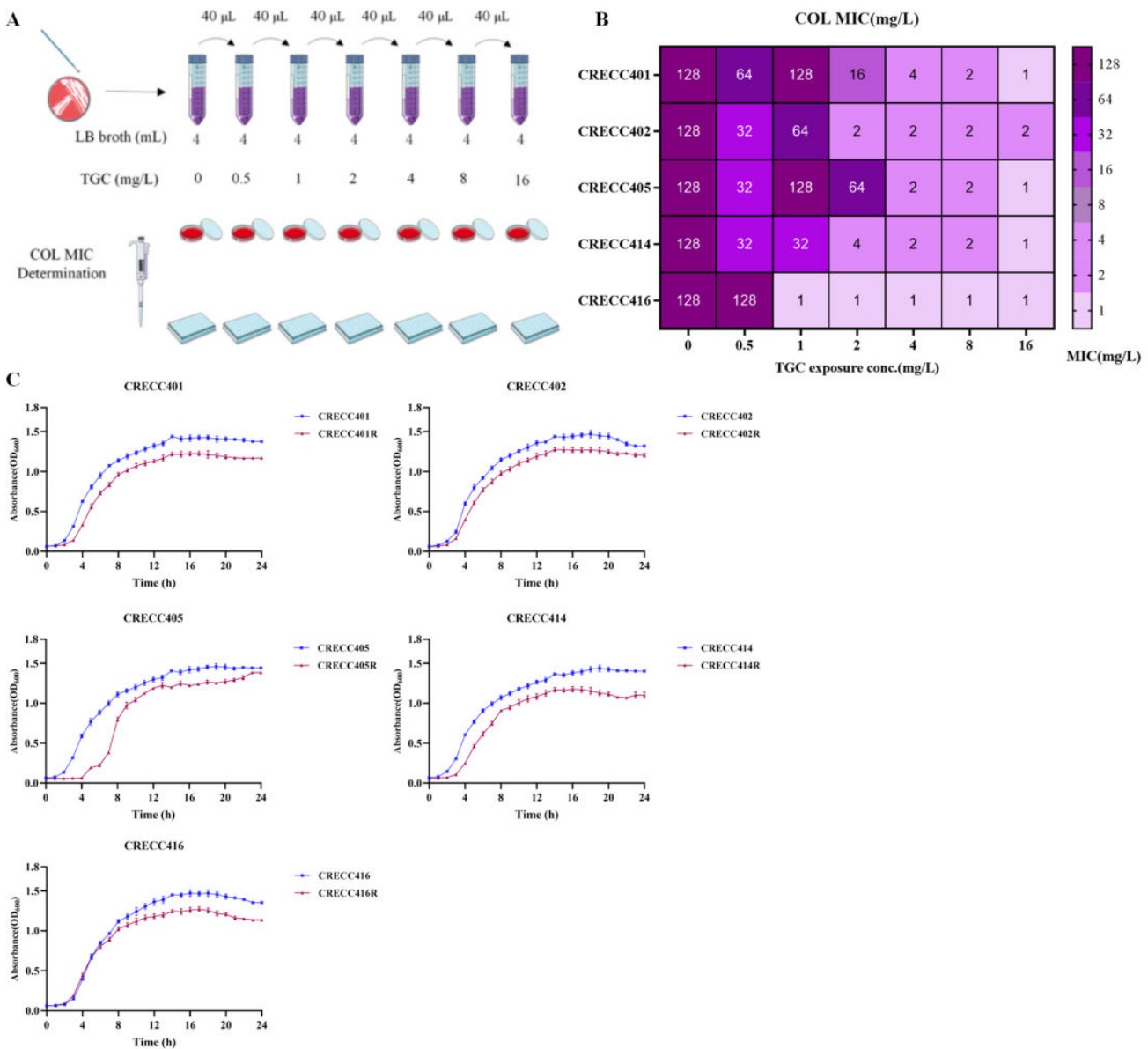

**FIG 2** Induction test and fitness cost assessment. (A) Schematic diagram of in *vitro* induction test. (B) Heat map of minimum inhibitory concentration (MIC) of colistin in different strains. (C) Growth curves of five strains of CRECC and their resistant descendants. COL, colistin; TGC, tigecycline.

CRECC416R and increased in CRECC402R and CRECC405R, without overall significance (Fig. 3). Among lipid A synthesis genes, *lpxC* and *lpxL* were highly expressed in some TGC-resistant mutants; however, the expression of *lpxA, lpxB, lpxC, lpxD*, and *lpxL* was not uniformly altered (Fig. 3).

## Fitness cost and stability analysis

To determine if TGC resistance mutations incur a fitness cost and assess stability, we evaluated the stability of TGC-resistant strains in antibiotic-free media (Fig. 4) and monitored the 24 h growth dynamics of CRTS strains and TGC-resistant strains (Fig. 2C and 4). After 10 days of passage, the MICs of tigecycline in all CRECC isolates decreased, with significant reductions observed for CRECC402R, CRECC405R, and CRECC414R. However, CRECC401R and CRECC416R retained their resistance to tigecycline. This resistance is attributed to a stable mutation in the *ramR* gene. CRECC401R and

**TABLE 2** Mutations of colistin and tigecycline resistance genes in five clinical isolates and five resistance-induced descendant strains[a,b]

| Isolates | Genes | | | | | | | | | |
|---|---|---|---|---|---|---|---|---|---|---|
| | phoP | phoQ | mgrB | acrA | acrB | marA | soxS | oqxB | ramA | ramR |
| CRECC401 | – | – | – | – | – | – | – | – | – | – |
| CRECC401R | – | – | – | – | – | – | – | – | – | Stop185 |
| CRECC402 | – | – | – | – | – | – | – | – | – | – |
| CRECC402R | – | – | – | – | – | – | – | – | – | – |
| CRECC405 | – | – | – | – | – | – | – | – | – | – |
| CRECC405R | – | – | – | – | – | – | – | – | – | – |
| CRECC414 | – | – | – | – | – | – | – | – | – | Frameshift |
| CRECC414R | – | – | – | – | – | – | – | – | W37K | Frameshift |
| CRECC416 | – | – | – | – | – | – | – | – | – | – |
| CRECC416R | – | – | – | – | – | – | – | – | – | IS26 |

[a]TGC resistance mutants CRECC401R, CRECC414R, and CRECC416R exist with premature termination, frameshift mutation, and insertion mutation.

[b]–, indicates that there is no mutation of the corresponding gene in all the strains in the study.

CRECC416R mutants shifted from induced susceptibility to COL resistance without restoring initial MIC levels, potentially due to evolutionary trade-offs and fitness costs after acquiring TGC resistance, despite no significant growth rate differences (51). Other mutants remained susceptible (Fig. 4). The TGC-resistant progeny did not exhibit a fitness cost compared with their parental strains ($P > 0.05$) and showed unstable resistance.

## Evaluation of combined efficacy of tigecycline and colistin against highly colistin-resistant isolates

A time-killing experiment was conducted on five CRTS strains using tigecycline (TGC) and COL, either individually or in combination. TGC was administered at a dose of 2 mg/L, and COL was given at 1/16 MIC (8 mg/L). COL alone initially inhibited growth against the five CRECC strains within 2 h; however, rapid regrowth ensued, reaching levels comparable with the antibiotic-free control group by 12 h (Fig. 5). TGC alone did not exert significant inhibitory effects up to 24 h; nevertheless, it maintained strain proliferation at this concentration, with all strains showing stable growth between 5 and 6 $\log_{10}$ CFU/mL. The combined treatment of TGC and COL demonstrated a bactericidal effect against CRECC, as evidenced by a decreasing trend in strain growth concentration within 24 h.

## DISCUSSION

There are few studies on ECC collateral sensitivity, leaving little available under the selective pressure of antibiotics (52). Therefore, identifying and confirming the collateral sensitivity of TGC in colistin-resistant CRECC could provide promise for the development of new clinical strategies.

RamA can directly regulate multidrug resistance efflux pumps AcrAB and OqxAB in *E. bugandensis* (53). In our study, we examined the molecular determinants of drug resistance in both parental and TGC-resistant mutants. A mutation in the transcription factor *ramR* led to a loss of regulatory function, resulting in upregulated expression of *ramA* and efflux pump genes *acrA* and *acrB*. This dysregulation likely explains the emergence of TGC-resistant mutants observed in this study, which is consistent with findings from other studies that attribute TGC resistance in *E. cloacae* to RamA-mediated overexpression of the AcrAB efflux pump (53). Additionally, we observed changes in the expression of *phoQ* (0.1-fold to 0.9-fold) and *arnA* (0.1-fold to 0.5-fold), with almost all TGC-resistant strains exhibiting lower levels compared with parental strains. The colistin resistance mechanisms of the ECC are complex and diverse, mainly including the overexpression of efflux pump genes (such as *acrA* and *acrB*), the modification of lipid A

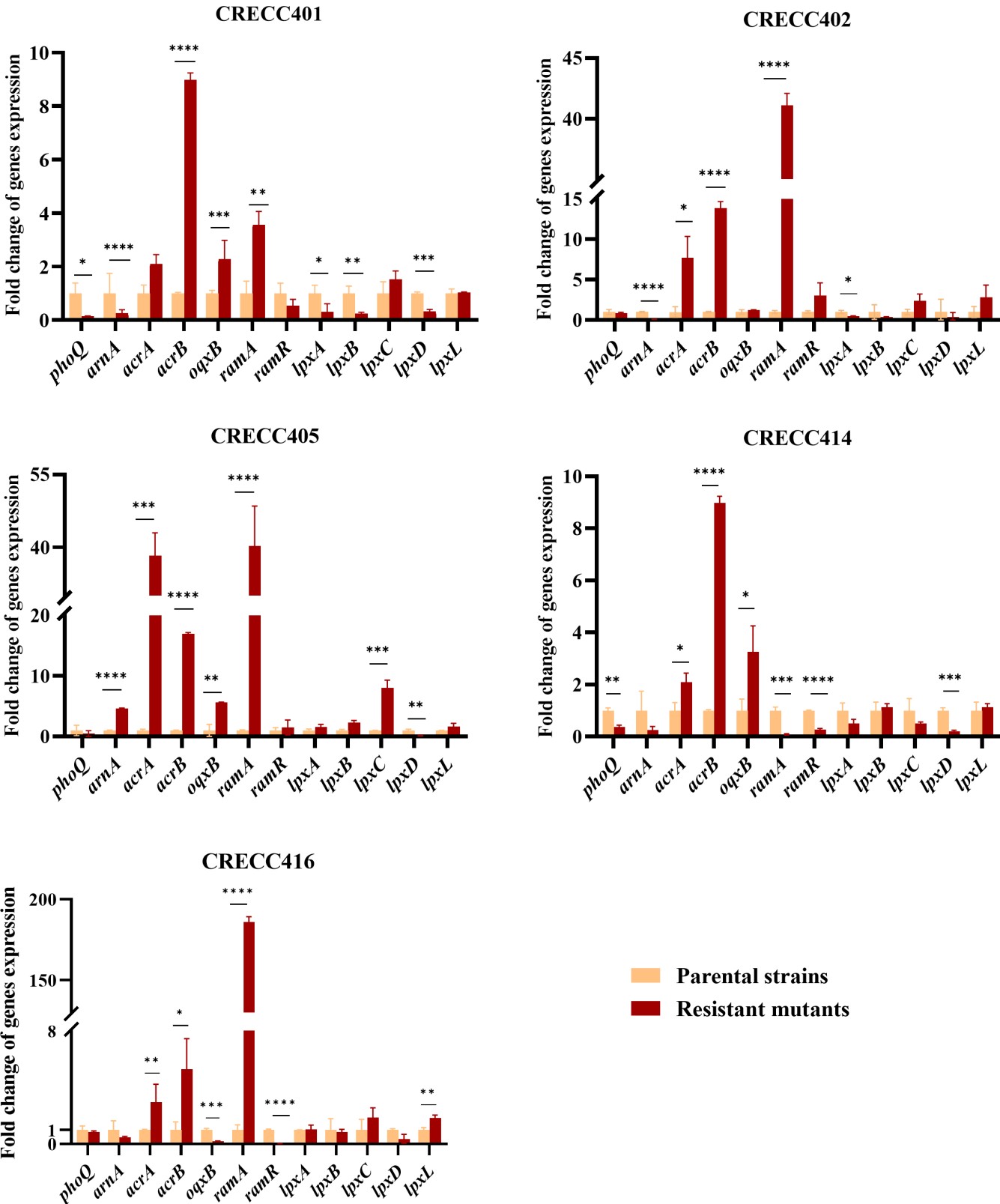

**FIG 3** Expression levels of genes associated with tigecycline and colistin resistance. $P < 0.05$; $P < 0.01$; $P < 0.001$; $P < 0.0001$.

(such as the addition of pETN and -L-Ara4N), and the mutations and overexpression of two-component regulatory systems (such as PhoPQ and PmrAB) (54). The decreased

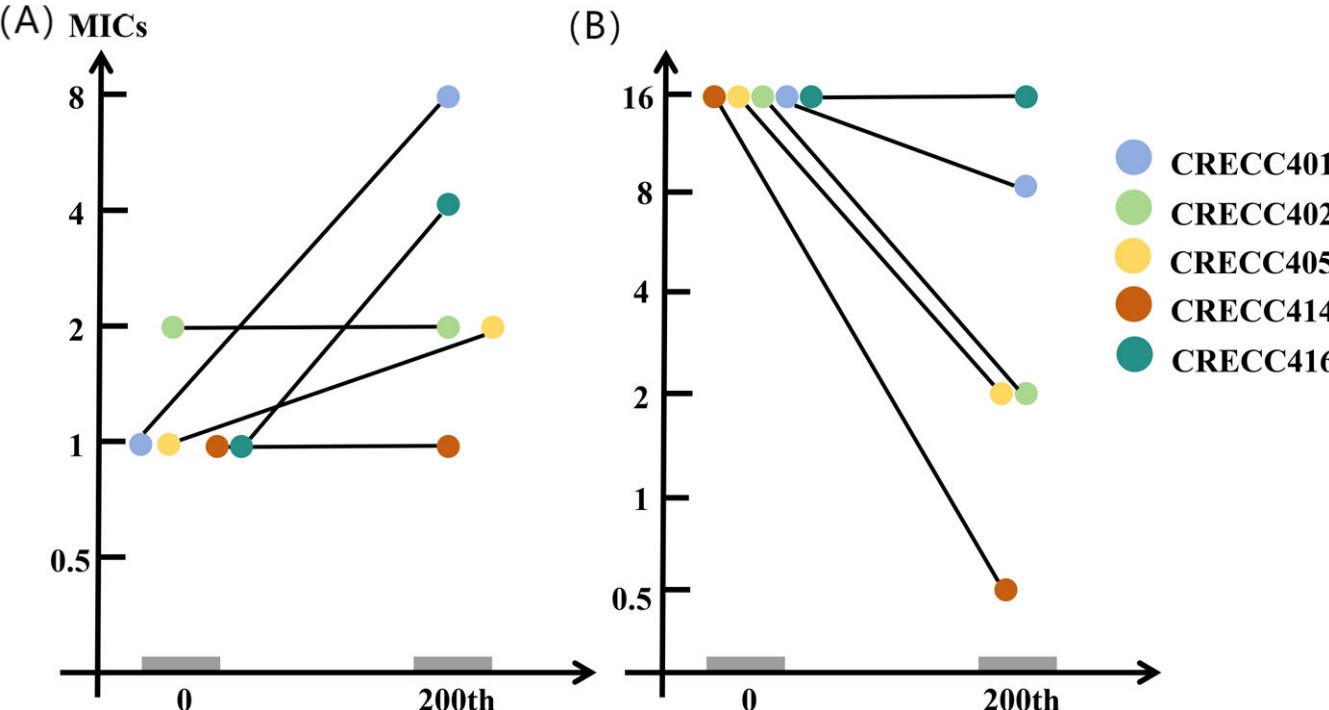

FIG 4 Stability test of tigecycline-resistant mutant strains. (A) Changes in colistin MICs. (B) Changes in tigecycline MICs. The X-axis shows the number of generations. When the tigecycline or colistin MICs of the five parental strains or of the five tigecycline-resistant mutants were identical, the figures are shown by one color.

expression of *phoQ* and *arnA* in TGC-resistant mutants, along with the disrupted expression of lipid A synthesis genes, likely accounts for the increased sensitivity to colistin observed in this study.

The colistin resistance pattern of the *Enterobacter cloacae* complex is closely associated with the presence of the *arnBCADTEF* operon (*arn*). The expression of *arn* is regulated by the two-component systems PmrAB and PhoPQ, as well as the PhoPQ feedback inhibitor MgrB. However, there may be additional unknown factors or regulatory pathways influencing the expression of *phoQ* and *arnA*, leading to inconsistencies in their expression levels among clinical strains (55). For example, in some strains (e.g., CRECC401R and CRECC402R), *arnA* is downregulated in isolates, whereas in others (e.g., CRECC405R), *arnA* is upregulated, despite sharing the same phenotype of decreased colistin resistance. In the CRECC402R strain, *phoQ* expression remained largely unchanged, whereas *arnA* expression was upregulated in the CRECC405R strain relative to the original strain. No mutations associated with colistin resistance were identified in CRECC402R, 405R, and 414R; the shift in their resistance may be related to the low expression of *phoQ* and *arnA*, as well as the inconsistent expression levels of lipid A synthesis genes compared with CRECC401R and CRECC416R. In the CRECC416R strain, the expression levels of *phoQ* and *arnA* did not decrease significantly; however, colistin sensitivity was restored. This could be because in an environment without colistin or other antibiotic selection pressures, the plasmid carrying the *mcr-9* gene is likely to be lost, as maintaining the plasmid may impose a fitness cost on the bacteria (56). Additionally, the modification of the lipid A portion of LPS can mediate RamA expression and lead to an increase in colistin sensitivity (57).

RamA modification regulates lipid A membrane biosynthesis at the permeability barrier and regulates critical multidrug efflux pump genes, such as AcrAB and OqxAB pumps, in *Enterobacterial* species (58). Increased *ramA* expression due to *ramR* deregulation is the primary mediator of tigecycline resistance in clinical isolates of *E. cloacae* and *E. aerogenes* (59). The mutation of the *ramR* gene inactivates the product RamR, which

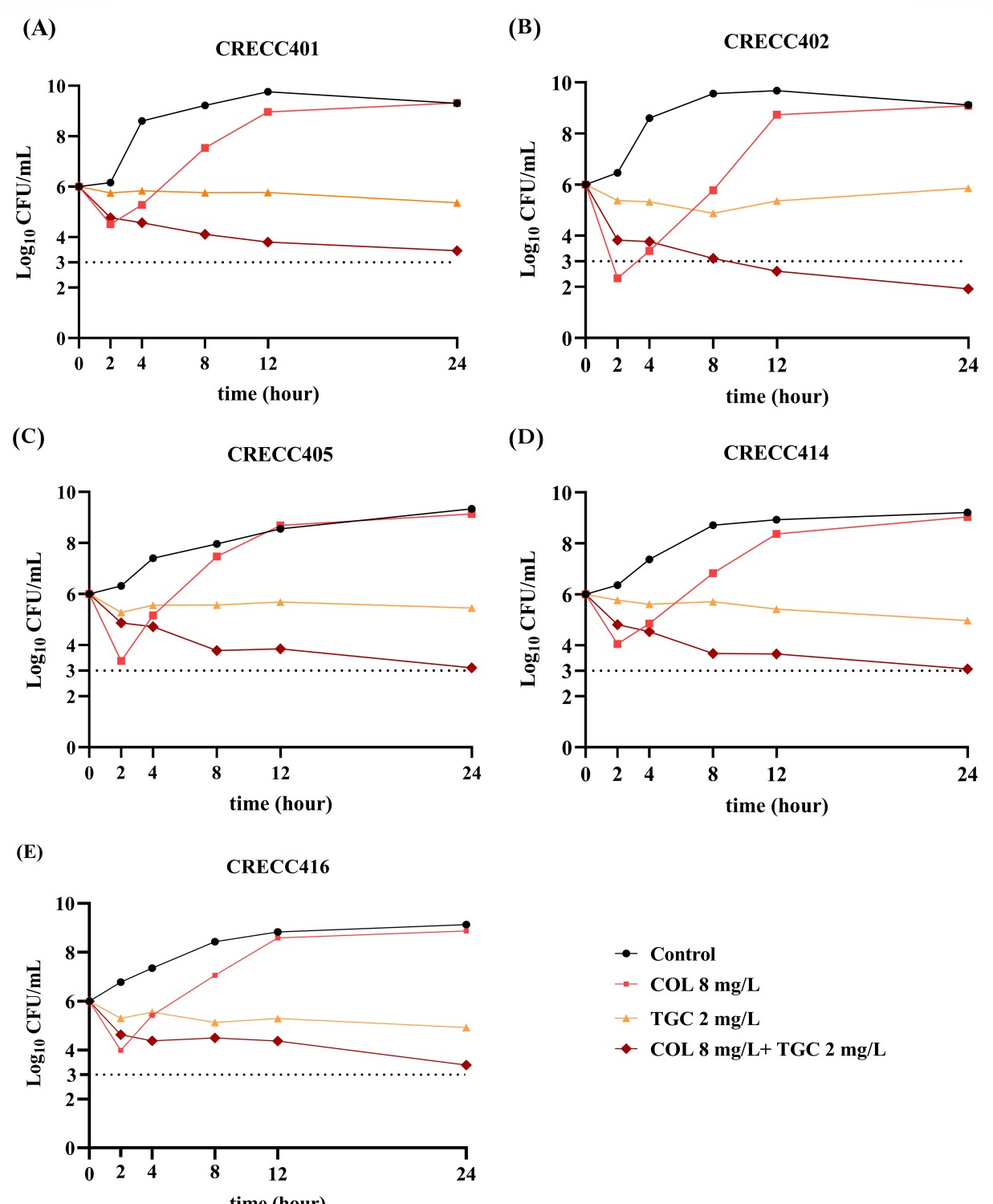

**FIG 5** Time-killing curve of 5 CRECC strains: The combination of TGC and colistin exhibits bactericidal activity. (A) to (E) are the bactericidal curve graphs corresponding to five bacterial strains, respectively. COL = colistin; TGC = tigecycline.

may lead to upregulation of *ramA* expression and TGC resistance. Consistent with this, transcription levels of the *ramA* gene increased significantly in CRECC401R and CRECC416R. The *ramA* gene of CRECC414R is not found to be upregulated; we speculate

that the missense mutation in *ramA* prevents its expression level, and this missense mutation may confer drug resistance to TGC. However, the expression of *acrA* and *acrB* genes in CRECC414R is upregulated. In a previous study, one *E. cloacae* isolate did not exhibit a corresponding decrease in *acrA* expression despite reduced *ramA* levels, and the tigecycline MICs for both the complemented and parental strains remained unchanged. This isolate also showed overexpression of *rarA* and *oqxA* (60). Notably, IS*26* is inserted after 127 base pairs of the *ramR* gene in CRECC416R. Similarly, in CRECC414 and CRECC414R, frameshift mutations occur due to the insertion of a single base between base pairs 126 and 127 of the *ramR* gene, and this region of the *ramA* gene sequence may be a popular mutation site. To trace the origin of IS*26* in the destructive *ramR* of CRECCC416R, we looked for the IS*26* element carried by the CRECC416 genome through comparative genomics. The results showed that the genome of CRECC416 carried four IS*26* elements, including two on the chromosome and two on the plasmid pA-CRECL416. To determine whether the IS*26* inserted into the *ramR* gene in CRECC416R came from a chromosome or a plasmid, we compared the IS*26* sequence in *ramR* with all four IS*26* elements across the genome. The results showed that the IS*26* element inserted in the *ramR* of CRECC416R had 100% homology with the four IS*26*s in the genome. Therefore, both chromosome and plasmid IS*26* elements may be transferred to the *ramR* gene of *E. cloacae* when exposed to TGC. CRECC402R and CRECC405R did not have any mutations in the *ramR* gene, and their *ramA* gene expression levels were still significantly increased. One study reported *E. cloacae* isolates showing elevated levels of *ramA* expression without any associated changes in *ramR*, suggesting the presence of multiple *ramA* regulatory pathways (59). Research has indicated the intergenic region (IR) between *ramR* and *ramA* is vital for regulating *ramA* expression and bacterial multidrug resistance. Specific IR sequences are crucial for RamR binding; mutations (e.g., substitutions, deletions) disrupt this binding, causing *ramA* overexpression and efflux pump activation, enhancing resistance. This is prominent in clinically isolated multidrug-resistant *K. pneumoniae*, *E. cloacae*, and *Salmonella* (61). Moreover, *ramR* mutations reduce its affinity for the *ramA* promoter (57).

Previous studies have shown that *ramA* can directly bind to and activate the expression of the promoters of the *lpxC, lpxL-2*, and *lpxO* genes (55, 59, 62). It has been reported that the loss of LPS results in colistin resistance in *Acinetobacter baumannii*. The mutations in lipid A biosynthesis genes, *lpxA, lpxC,* and *lpxD*, cause total loss of LPS production, halting colistin binding to the membrane and hence colistin resistance (63). Colistin-resistant *K. pneumoniae* was found to exhibit multiple resistance mechanisms. Among the genetic mutations identified, mutations in *lpxA, lpxB, lpxD*, and *lpxM* were observed. Additionally, the LPS export gene *lptD* showed a deleterious mutation (64). Faulty lipid A and defective LPS assembly may have contributed to colistin resistance, as demonstrated previously. Our results showed that there was no significant consistency in the expression of lipid synthesis genes, and the expression of *lpxA, lpxB, lpxC, lpxD,* and *lpxL* genes was dislocated among the five CRECC strains and their drug-resistant mutants. Disturbed expression of lipid A synthesis genes in the present study may also be associated with altered colistin resistance. The mechanism of resistance to both colistin and TGC involves the synthesis of lipids. We speculate that these strains with high levels of colistin resistance in response to TGC exposure have difficulty in their own lipid A synthesis and modification to cope with the effects of both drugs (65).

The concept of CS in treatment design offers a promising approach to curb the reemergence of infections by suppressing or even reversing resistance evolution (33). Leveraging CS, such as the sensitivity of colistin-resistant *Enterobacter cloacae* complex to TGC, in treatment strategies could involve antibiotic combination therapy or a multi-drug approach, potentially reducing treatment times and avoiding resistance development (66, 67). Ensuring the persistence of CS across bacterial strains with varying mutational backgrounds is crucial for its effective translation into clinical practice. This is particularly important when investigating the resilience of CS in previously established antibiotic-resistant mutants exposed to novel antimicrobials, a situation that has

not been extensively studied in a clinical context (68). CS-based therapies hold the greatest promise for delivering personalized infectious disease treatments, especially for antibiotics with a limited therapeutic window (37).

This study has limitations. First, the inclusion of a limited number of clinical isolates restricts the applicability of our findings to clinical drug-resistant strains. Second, the hypothesis that TGC induces colistin resistance by interfering with lipid synthesis is speculative, as it lacks supporting lipidomic or proteomic data.

## Conclusion

In summary, this study found that after clinical isolates of colistin-resistant CRECC were subjected to the selective pressure of tigecycline. Tigecycline resistance was mainly due to the overexpression of *ramA*, leading to an increased level of the AcrAB-TolC efflux pump. The reduced sensitivity to colistin may be due to the disorganization of lipid synthesis caused by *ramA*. The recovery of colistin resistance in tigecycline-resistant mutants after the removal of antibiotic pressure also confirmed this point. In *vitro* combination of the two antibiotics is superior to their individual use.

## ACKNOWLEDGMENTS

This work was supported by the Joint projectProject of Chongqing Health Commission and Science and Technology Bureau (2023MSXM018), Yongchuan Natural Science Foundation (2021yc-jckx20053), Talent introduction projectIntroduction Project of Yongchuan Hospital of Chongqing Medical University (YJYJ202004 and YJYJ202005), and Program for Youth Innovation in Future Medicine and Chongqing Medical University (W0113).

K.Y. and J.W. contributed equally to this work.

K.Y. was primarily responsible for experimental operations and data analysis, while J.W. was in charge of article writing and manuscript revision.

M.L. and J.W. contributed equally to this work. K.Y., J.W., J.W., C.L., and X.Z. designed and coordinated the the study. X.Y. and J.L. participated in the whole experiment process. J.W. drafted the article. C.W., J.W., and X.L. helped with the experimental process. J.W., W.L., X.G., and Y.Y. provided the samples and the clinical data. J.W. and K.W. analyzed and interpreted the data.

All authors contributed to the article and approved the submitted version.

## AUTHOR AFFILIATIONS

[1]Department of Microbiology, Yongchuan Hospital of Chongqing Medical University, Chongqing, China
[2]Department of Pathogenic Biology, Basic Medicine of Jiamusi University, Jiamusi, China
[3]Department of Life Science and Technology, Mudanjiang Normal University, Mudanjiang, China

## AUTHOR ORCIDs

Kaixin Yu http://orcid.org/0009-0007-1776-1649
Chunjiang Li http://orcid.org/0009-0004-6446-4370
Xiaoli Zhang http://orcid.org/0000-0002-9385-8107

## AUTHOR CONTRIBUTIONS

Kaixin Yu, Conceptualization, Formal analysis, Software, Writing – review and editing | Jiming Wu, Formal analysis, Validation, Visualization, Writing – original draft | Mingjing Liao, Resources, Supervision | Jianmin Wang, Methodology, Resources | Chunli Wei, Data curation, Investigation | Wenzhang Long, Visualization | Xuemei Gou, Investigation | Yang Yang, Formal analysis | Jin Wang, Investigation | Xushan Liang, Formal analy-

sis | Chunjiang Li, Project administration | Xiaoli Zhang, Funding acquisition, Project administration

## DATA AVAILABILITY

The data sets generated during and/or analyzed during the current study are available in this manuscript.

The genome sequence has been submitted to the National Center for Biotechnology Information, and the accession number is provided in the Table S1. All primers used in this study are shown in the Table S2.

## ADDITIONAL FILES

The following material is available online.

### Supplemental Material

**Supplemental tables (Spectrum03310-24-s0001.docx).** Table S1, clinical isolates of carbapenem-resistant *Enterobacter cloacae* complex (CRECC) with colistin resistance; Table S2, primers used in this study.

### Open Peer Review

**PEER REVIEW HISTORY (review-history.pdf).** An accounting of the reviewer comments and feedback.

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
