## [Reviewer comments · Microbiology Spectrum]

Microbiology Spectrum

Study on collateral sensitivity of tigecycline to colistin resistant *Enterobacter cloacae* complex

Kaixin Yu, Jiming Wu, Mingjing Liao, Jianmin Wang, Chunli Wei, WenZhang Long, Xuemei Gou, Yang Yang, Jin Wang, Xushan Liang, Chunjiang Li, and Xiaoli Zhang

Corresponding Author(s): Xiaoli Zhang, Yongchuan Hospital of Chongqing Medical University

Review Timeline:

Submission Date:	January 8, 2025
Editorial Decision:	March 4, 2025
Revision Received:	March 11, 2025
Editorial Decision:	April 7, 2025
Revision Received:	April 10, 2025
Editorial Decision:	April 15, 2025
Revision Received:	April 16, 2025
Accepted:	April 17, 2025

Editor: Siu-Kei Chow

Reviewer(s): Disclosure of reviewer identity is with reference to reviewer comments included in decision letter(s). The following individuals involved in review of your submission have agreed to reveal their identity: François Guerin (Reviewer #1)

Transaction Report:

DOI: <https://doi.org/10.1128/spectrum.03310-24>

Re: Spectrum03310-24 (**Study on collateral sensitivity of tigecycline to colistin resistant *Enterobacter cloacae* complex**)

Dear Dr. Xiaoli Zhang:

Thank you for the privilege of reviewing your work. Below you will find my comments, instructions from the Spectrum editorial office, and the reviewer comments.

Revision Guidelines

Sincerely,
Siu-Kei Chow
Editor
Microbiology Spectrum

Reviewer #1 (Comments for the Author):

Study on collateral sensitivity of tigecycline to colistin resistant *Enterobacter cloacae* complex

Authors: Kaixin Yu, Jiming Wu, Mingjing Liao, Jianmin Wang, Chunli Wei, Wenzhang Long, Xuemei Gou, Yang Yang, Jin Wang, Xushan Liang, Chunjiang Li, Xiaoli Zhang

The aim of this study was to investigate the collateral sensitivity of tigecycline to colistin in clinically isolated carbapenem-

resistant *Enterobacter cloacae* complex (CRECC). The in vitro efficacy of tigecycline and colistin alone or in combination was fully evaluated to provide new theoretical insights for the clinical treatment of CRECC.

General comments:

this article is interesting, but a lot of inaccuracies have been reported.

It would have been desirable to sequence the resistant TGC strains and compare the sequences with the sensitive TGC strains. In the discussion, it would be advisable to integrate the conclusions of the article doi: 10.2147/IDR.S473580 on the various mechanisms involved in colistin resistance.

Major comments:

Line 241: How do you explain CRECC402 strain's resistance to penemes? This strain seems to be losing its resistance plasmid?

Line 243: the strains (CRECC401 and CRECC402) do not possess the 2-component phoP/phoQ system? would the strains have deleted this system?

Line 241-250: How do you explain the resistance of these strains to colistin? do you have heteroresistance to colistin or homogeneous resistance?

Lines 252-263: How do you explain the fact that the CRECC402R strain has become sensitive to beta-lactams? Has it lost its plasmid conferring resistance?

Lines 267-271: no mutations were found in any of the genes mentioned (table 2), with the exception of ramR or ramA?

Line 280: the fold change observed is not quantified in the text. it would be advisable to revise figure 3 and use Log10 as the ordinate?

Line 320: *E. cloacae* reference missing (doi: 10.1128/AAC.00962-20)

Lines 320-327: This has already been noted in the following publication: doi: 10.1128/AAC.00962-20

Lines 327-329: fold changes are not indicated?

Minor comments:

- Line 65 and throughout the text: please write "in vitro" in italics

- Line 68 and throughout the text: please write "*Enterobacter*" in italics

- Line 101 and throughout the text: please replace "*Klebsiella pneumoniae*" by "*K. pneumoniae*"

- Line 242 and throughout the text: please replace "metal-beta lactamase gene" by "metallo-beta-lactamase gene"

- the abbreviations COL and TGC are not used throughout the text! please correct!

Reviewer #2 (Comments for the Author):

General comments: In their article entitled "Study on collateral sensitivity of tigecycline to colistin resistant *Enterobacter cloacae* complex" Kaixin Yu and colleagues explored the mechanism behind the sensitivity to colistin associated with the emergence of tigecycline resistance among five carbapenem resistant *Enterobacter cloacae* complex clinical isolates. The article is interesting, and explore how the development of resistance to one molecule can transform an initially resistant strain to sensitive considering another molecule. The author explored the duo "colistin - tigecycline". In general, English could be improved as some sentences were difficult to understand. Scientifically speaking, I had some questions/remarks which are detailed below.

Major comments:

Line 244: if the mcr variant was the mcr-9 which is not described as responsible for high colistin MICs, do you have an explanation about the original colistin MICs among the clinical isolates?

Line 255: was the decrease of the colistin MICs associated to a loss of plasmid carrying the mcr gene?

Lines 261 - 262: "The accessory sensitivity induced by TGC exposure was prominently accessory sensitivity induced by TGC exposure was prominently reflected in colistin, while changes in resistance to other antibiotic classes were inconsistent" when the MICs of quinolones did not change with the tigecycline resistance could it be explain by mutations within QRDR? Did you look at these mutations?

Line 286 [...] but was downregulated in CRECC414R, correlating with a missense mutation in ramA (Fig 3). How could you explain the up regulation of acrB if RamA was absent considering that no other regulator has been found upregulated?

Figure 4 is very difficult to read and understand due to the five strains were not all visible in every time points. The general idea is really good but need to be improve.

Figure 3: what is your explanation about the transcriptomic response of phoQ and arnA associated with tigecycline resistance?

What is the promoter of these genes? If they are known, it could be very interesting to study their transcriptomic response associated with tigecycline resistance.

Could you please discuss the fact that transcriptomic response of *phoQ* and *arnA* are not homogenous among the clinical strains i.e for some of them CRECC401, CRECC402 the gene *arnA* is down regulated among tigecycline resistant strains while in CRECC405 *arnA* is up regulated despite the same phenotype: decrease resistance to colistin.

Lines 327-329: "Additionally, we observed changes in the expression of *phoQ* and *arnA*, with almost all TGC-resistant strains exhibiting lower levels compared to parental strains." Almost all represented 3/5 strains, what about the two others?

Regarding the CRECC416, how could you explain the decreased sensitivity of colistin while there was no transcriptomic modification in *phoQ* and *arnA*?

Line 299 "Colistin resistance patterns shifted from sensitive to resistant or remained unchanged in these strains (Figure 4)." But the MICs of colistin were lower than the original strains MICs > 128 mg/L? could you please comment?

Times kill curves: Could you please add the number 3 in the y abscises in order to see the bactericidal effect of the combination

Lines 351 - 354 "Similarly, in CRECC414 and 352 CRECC414R, frameshift mutations occur due to the insertion of a single base between 353 base pairs 126 and 127 of the *ramR* gene, and this region of the *ramA* gene sequence 354 may be a popular mutation site" What are the consequences in the RamR protein of the frame shift mutations?

Line 363 - 368 "CRECC402R and 364 CRECC405R did not have any mutations in the *ramR* gene, and their *ramA* gene expression levels were still significantly increased, so there may be other relevant regulatory factors. Several studies have similarly reported *E. cloacae* isolates showing elevated levels of *ramA* expression without any associated changes in *ramR*, suggesting the presence of multiple *ramA* regulatory pathways(43)"

Gravey F et al. in Central role of the *ramAR* locus in the multidrug resistance in ESBL *Enterobacteriales*. *Microbiol Spectr* 12:e03548-23 and Baucheron S et al in Binding of the RamR Repressor to Wild-Type and Mutated Promoters of the *ramA* Gene Involved in Efflux-Mediated Multidrug Resistance in *Salmonella enterica* Serovar Typhimurium. *Antimicrob Agents Chemother* 56: have shown the importance of the intergenic region between *ramR* and *ramA*, could you look at any genomic mutations?

Lines 369 - 371 "Previous studies have shown that RamA can activate *lpxC*, *lpxL*, and *lpxO* associated with lipid A biosynthesis, while lipid A modification can lead to a variety of outcomes, such as reduced colistin sensitivity (44-46)." In this publication De Majumdar et al Elucidation of the RamA regulon in *Klebsiella pneumoniae* reveals a role in LPS regulation. *PLoS pathogens*, 11(1), e1004627. The increased of RamA is associated with increase of colistin MICs, could you please comment?

Minor comments:

Lines 62 - 63: please add references about « collateral sensitivity"

Lines 71 - 74: there are other new options against CRECC like aztreonam-avibactam or cefiderocol, could you please discuss these molecules?

Line 80: Are tetracycline and colistin really widespread in Human medicine? Line 82: could you please precise the variant of *mcr* which is associated to CRECC?

Lines 104 - 105 "CRECC poses greater complexity compared to carbapenem--resistant *Klebsiella pneumoniae*, further research in this area remains limited." Could you please explain why ?

Lines 163 - 164: "All strains before and after induction were detected by polymerase chain reaction (PCR) and sanger sequencing." I don't understand this sentence

Lines 197 - 200: could you please explain the relationship between stability of MICs and growth curves?

Lines 213: in the Time Kill Curve part, could you please explain the dilution ration of 1:100?

Line 235: what was the method used to assess the genomic distance between the five strains? Could you please provide the Hoffman *hsp60* cluster to your five strains?

Line 239: These are not the epidemic sequence types of ECC infection in China or our hospital. Could you please cite the species and the MLST of the epidemic strains?

Line 298 "However, CRECC401R and CRECC416R maintained their resistance to tigecycline" due to fixed ramR mutations, this should be mentioned.

Line 317 - 319 "Consequently, uncovering and demonstrating the collateral sensitivity of TGC in colistin-resistant CRECC holds immense potential for novel clinical interventions." Could you please be more prudent about this sentence and rephrase? Thank you.

Line 320 -321: "RamA can directly regulate multidrug resistance efflux pumps AcrAB and OqxAB in 321 K. pneumoniae(39) » The effect of RamA on AcrAB have also been published in Ecc strains, could you please add appropriate references.

General comments:

In their article entitled “*Study on collateral sensitivity of tigecycline to colistin resistant Enterobacter cloacae complex*” Kaixin Yu and colleagues explored the mechanism behind the sensitivity to colistin associated with the emergence of tigecycline resistance among five carbapenem resistant *Enterobacter cloacae* complex clinical isolates.

The article is interesting, and explore how the development of resistance to one molecule can transform an initially resistant strain to sensitive considering another molecule. The author explored the duo “colistin – tigecycline”.

In general, English could be improved as some sentences were difficult to understand. Scientifically speaking, I had some questions/remarks which are detailed below.

Major comments:

Line 244: if the *mcr* variant was the *mcr-9* which is not described as responsible for high colistin MICs, do you have an explanation about the original colistin MICs among the clinical isolates?

Line 255: was the decrease of the colistin MICs associated to a loss of plasmid carrying the *mcr* gene?

Lines 261 – 262: “*The accessory sensitivity induced by TGC exposure was prominently accessory sensitivity induced by TGC exposure was prominently reflected in colistin, while changes in resistance to other antibiotic classes were inconsistent*” when the MICs of quinolones did not change with the tigecycline resistance could it be explain by mutations within QRDR? Didi you looked at these mutations?

Line 286 [...] but was downregulated in CRECC414R, correlating with a missense mutation in *ramA* (Fig 3). How could you explain the up regulation of *acrB* if RamA was absent considering that no other regulator has been found upregulated?

Figure 4 is very difficult to read and understand due to the five strains were not all visible in every time points. The general idea is really good but need to be improve.

Figure 3: what is your explanation about the transcriptomic response of *phoQ* and *arnA* associated with tigecycline resistance? What is the promoter of these genes? If they are known, it could be very interesting to study their transcriptomic response associated with tigecycline resistance.

Could you please discuss the fact that transcriptomic response of *phoQ* and *arnA* are not homogenous among the clinical strains i.e for some of them CRECC401, CRECC402 the gene *arnA* is down regulated among tigecycline resistant strains while in CRECC405 *arnA* is up regulated despite the same phenotype: decrease resistance to colistin.

Lines 327-329: “*Additionally, we observed changes in the expression of phoQ and arnA, with almost all TGC-resistant strains exhibiting lower levels compared to parental strains.*” Almost all represented 3/5 strains, what about the two others?

Regarding the CRECC416, how could you explain the decreased sensitivity of colistin while there was no transcriptomic modification in *phoQ* and *arnA*?

Line 299 “*Colistin resistance patterns shifted from sensitive to resistant or remained unchanged in these strains (Figure 4).*” But the MICs of colistin were lower than the original strains MICs > 128 mg/L? could you please comment?

Times kill curves: Could you please add the number 3 in the y abscises in order to see the bactericidal effect of the combination

Lines 351 – 354 “*Similarly, in CRECC414 and 352 CRECC414R, frameshift mutations occur due to the insertion of a single base between 353 base pairs 126 and 127 of the ramR gene, and this region of the ramA gene sequence 354 may be a popular mutation site*” What are the consequences in the RamR protein of the frame shift mutations?

Line 363 - 368 “*CRECC402R and 364 CRECC405R did not have any mutations in the ramR gene, and their ramA gene expression levels were still significantly increased, so there may be other relevant regulatory factors. Several studies have similarly reported E. cloacae isolates showing elevated levels of ramA expression without any associated changes in ramR, suggesting the presence of multiple ramA regulatory pathways(43)*”

Gravey F *et al.* in *Central role of the ramAR locus in the multidrug resistance in ESBL-Enterobacteriales*. Microbiol Spectr 12:e03548-23 and Baucheron S *et al* in *Binding of the RamR Repressor to Wild-Type and Mutated Promoters of the ramA Gene Involved in Efflux-Mediated Multidrug Resistance in Salmonella enterica Serovar Typhimurium*. Antimicrob Agents Chemother 56: have shown the importance of the intergenic region between *ramR* and *ramA*, could you look at any genomic mutations?

Lines 369 – 371 “*Previous studies have shown that RamA can activate lpxC, lpxL, and lpxO associated with lipid A biosynthesis, while lipid A modification can lead to a variety of outcomes, such as reduced colistin sensitivity (44-46).*” In this publication De Majumdar *et al* *Elucidation of the RamA regulon in Klebsiella pneumoniae reveals a role in LPS regulation. PLoS pathogens, 11(1), e1004627*. The increased of RamA is associated with increase of colistin MICs, could you please comment?

Minor comments:

Lines 62 – 63: please add references about « collateral sensitivity »

Lines 71 – 74: there are other new options against CRECC like aztreonam-avibactam or cefiderocol, could you please discuss these molecules?

Line 80: Are tetracycline and colistin really widespread in Human medicine?

Line 82: could you please precise the variant of *mcr* which is associated to CRECC?

Lines 104 – 105 “*CRECC poses greater complexity compared to carbapenem--resistant Klebsiella pneumoniae, further research in this area remains limited.*” Could you please explain why ?

Lines 163 – 164: “*All strains before and after induction were detected by polymerase chain reaction (PCR) and sanger sequencing.*” I don’t understand this sentence

Lines 197 – 200: could you please explain the relationship between stability of MICs and growth curves?

Lines 213: in the Time Kill Curve part, could you please explain the dilution ration of 1:100?

Line 235: what was the method used to assess the genomic distance between the five strains?
Could you please provide the Hoffman *hsp60* cluster to your five strains?

Line 239: These are not the epidemic sequence types of ECC infection in China or our hospital.
Could you please cite the species and the MLST of the epidemic strains?

Line 298 “*However, CRECC401R and CRECC416R maintained their resistance to tigecycline*”
due to fixed ramR mutations, this should be mentioned.

Line 317 – 319 “*Consequently, uncovering and demonstrating the collateral sensitivity of TGC in colistin-resistant CRECC holds immense potential for novel clinical interventions.*” Could you please be more prudent about this sentence and rephrase? Thank you.

Line 320 -321: “*RamA can directly regulate multidrug resistance efflux pumps AcrAB and OqxAB in 321 K. pneumoniae(39)* » The effect of RamA on AcrAB have also been published in Ecc strains, could you please add appropriate references.

Response Letter

Dear editors and reviewers,

Manuscript number: Spectrum03310-24

Title: Study on collateral sensitivity of tigecycline to colistin resistant *Enterobacter cloacae* complex

We appreciate the opportunity to revise our manuscript titled "Study on collateral sensitivity of tigecycline to colistin resistant *Enterobacter cloacae* complex" and are grateful for the insightful comments provided by the reviewers. Those comments are all valuable and very helpful for revising and improving our paper, as well as the important guiding significance to our researches. In the following, we have provided detailed responses to each of the reviewers' comments. Revised portions are marked in red in the paper. Additionally, we have conducted a comprehensive revision of the entire manuscript. In this response letter, the reviewers' comments are presented in italicized font and special concerns have been numbered. Our corresponding changes and additions to the manuscript are highlighted in red text in "Marked-Up_Manuscript". The response is given in normal font and changes. We have tried our best to make all the revisions clear, and we hope that the revised manuscript meets the requirements for publication.

Response to reviewer #1 Comments

The aim of this study was to investigate the collateral sensitivity of tigecycline to colistin in clinically isolated carbapenem-resistant Enterobacter cloacae complex (CRECC). The in vitro efficacy of tigecycline and colistin alone or in combination was fully evaluated to provide new theoretical insights for the clinical treatment of CRECC.

Overall response to Reviewer 1: Thank you for spending time in reviewing our manuscript and providing us with a list of constructive comments.

General comments:

this article is interesting, but a lot of inaccuracies have been reported. It would have been desirable to sequence the resistant TGC strains and compare the sequences with the sensitive TGC strains. In the discussion, it would be advisable to integrate the conclusions of the

article doi: 10.2147/IDR.S473580 on the various mechanisms involved in colistin resistance.

The author's answer: We would like to express our sincere gratitude to the reviewer for your insightful comments and valuable feedback.

Sequencing the TGC-resistant strains and comparing their sequences with those of TGC-susceptible strains is indeed a meaningful endeavor. This paper focuses specifically on identifying the reasons for the restoration of colistin sensitivity in tigecycline-resistant mutants. Therefore, we only determined the mutations in specific genes, such as efflux pump genes and their corresponding regulatory factors associated with tigecycline resistance, as well as colistin resistance genes. As a result, we overlooked this aspect. Our research team will continue to improve upon this in our future work. Thank you for your constructive suggestion.

We sincerely appreciate the valuable comments. We have checked the literature carefully and added this reference on various mechanisms involved in colistin resistance into the Discussion part in the revised manuscript.

The added text and line numbers 276 - 381:

The colistin resistance mechanisms of the ECC are complex and diverse, mainly including the overexpression of efflux pump genes (such as *acrA* and *acrB*), the modification of lipid A (such as the addition of pETN and -L-Ara4N), and the mutations and overexpression of two-component regulatory systems (such as *PhoPQ* and *PmrAB*)(54).

Major comments:

1. Line 241: How do you explain CRECC402 strain's resistance to penemes? This strain seems to be losing its resistance plasmid?

Thank you for your insightful suggestion. I'm not quite sure whether your question is about the parent strain or the mutant strain, so I've provided explanations for both.

The study of bacterial collateral sensitivity is still in its infancy. The heterogeneity of collateral sensitivity effects due to the stochasticity of evolutionary trajectories(doi: 10.1093/molbev/msx158), low reproducibility, and factors such as epigenetic effects, gene deletion, and horizontal gene transfer (doi: 10.1038/s41467-018-08098-6) in bacteria all increase the complexity and unpredictability of bacterial resistance evolution in real-world scenarios. The reasons for the decreased resistance to carbapenems and changes in sensitivity to other antibiotic classes in the TGR mutant CRECC402R may be similar.

Although CRECC402 possesses only the carbapenemase gene IMP-4, blaIMP-4 has established low levels of endemicity in many regions. Imipenemase metallo-β-lactamase types alone accounted for more than half of all carbapenemases in carbapenemase-producing Enterobacterales isolates in Australia, particularly in the Enterobacter cloacae complex (DOI: 10.1111/imj.16374).

Lines 174-180:

Additionally, the high-level COL resistance observed inCRECC401andCRECC402clinical isolates may be attributed to other mechanisms, such as the overexpression of efflux pumps (e.g.,AcrAB-TolC) and biofilm formation.*Enterobacter cloacae* can expel COL from cells via efflux pump systems, reducing its intracellular accumulation and conferring resistance . Overexpression of regulatory genes such as*soxS*and*ramA*can further enhance resistance by activating efflux pump systems(46).

Reference:

*2. Line 243: the strains (CRECC401 and CRECC402) do not possess the 2-component phoP/phoQ system? would the strains have deleted this system?*

Thank you for your insightful question. CRECC401 and CRECC402 possess the PhoP/PhoQ two-component system, and these strains have not deleted this system. Variations in gene expression levels can be observed in Figure 3 of the study results.

In the revised manuscript, the corresponding explanation has been looked in Figure 1. As follows:

Fig 1. Clustering analysis and resistance determinants of 5 CRECC strains from our institution. The resulting structure reflected the similarity between the sequences, and different resistance determinants present (in different color) in each strain are shown on the right.

Fig 3. Expression levels of genes associated with tigecycline and colistin resistance. *P < 0.05; **P < 0.01; ***P < 0.001; ****P < 0.0001.

3. Line 241-250: How do you explain the resistance of these strains to colistin? do you have heteroresistance to colistin or homogeneous resistance?

Thank you for your insightful question. Unfortunately, we did not find mutations in *PmrAB*, *PhoPQ*, or *mgrB* in the strains. This could be due to the overexpression of *PmrAB* and *PhoPQ*, which may lead to the modification of lipid A through the addition of phosphoethanolamine (pETN) and/or 4-amino-4-deoxy-l-arabinose (-L-Ara4N).

The reasons for the resistance of these strains to polymyxin have been supplemented in lines

166 - 180 of the manuscript.

In our previous study, the *mcr-9* gene in CRECC402 and CRECC405 strains was located on the IncHI2/2A plasmid, with a genetic context of *IS903B-mcr-9-wbuC-IS26*. In CRECC414, the *mcr-9* gene was found on the pECL414-1 plasmid, flanked by *IS1B-mcr-9.2-wbuC-IS26* (11). In CRECC416, the *mcr-9.1* gene was located upstream of *IS903B* and downstream of *IS481* and *IS26* (44). These insertion sequences may facilitate the stable expression and dissemination of the *mcr-9* gene, potentially serving as key factors in the spread of COL resistance among CRECC strains in hospital settings. Additionally, the high-level COL resistance observed in CRECC401 and CRECC402 clinical isolates may be attributed to other mechanisms, such as the overexpression of efflux pumps (e.g., AcrAB-TolC) and biofilm formation. *Enterobacter cloacae* can expel COL from cells via efflux pump systems, reducing its intracellular accumulation and conferring resistance (45). Overexpression of regulatory genes such as *soxS* and *sdra* can further enhance resistance by activating efflux pump systems (46).

References:

- [11] Jiang S, Wang X, Yu H, Zhang J, Wang J, Li J, Li X, Hu K, Gong X, Gou X, Yang Y, Li C, Zhang X. 2022. Molecular antibiotic resistance mechanisms and co-transmission of the *mcr-9* and metallo- β -lactamase genes in carbapenem-resistant *Enterobacter cloacae* complex. *Front Microbiol* 13:1032833.
- [44] Li W, Zhang J, Fu Y, Wang J, Liu L, Long W, Yu K, Li X, Wei C, Liang X, Wang J, Li C, Zhang X. 2024. In vitro and in vivo activity of ceftazidime/avibactam and aztreonam alone or in combination against *mcr-9*, serine- and metallo- β -lactamases-co-producing carbapenem-resistant *Enterobacter cloacae* complex. *European Journal of Clinical Microbiology & Infectious Diseases* 43:1309-1318.
- [45] Mousavi SMJ, Hosseinpour M, Rafiei F, Mahmoudi M, Shahraki H, Shiri H, Hashemi A, Sharahi JY. 2025. Colistin antibacterial activity, clinical effectiveness, and mechanisms of intrinsic and acquired resistance. *Microb Pathog* doi:10.1016/j.micpath.2025.107317:107317.
- [46] Telke AA, Olaitan AO, Morand S, Rolain JM. 2017. *soxRS* induces colistin hetero-resistance in *Enterobacter asburiae* and *Enterobacter cloacae* by regulating the *acrAB-tolC* efflux pump. *J Antimicrob Chemother* 72:2715-2721.

The explanation of whether *Pseudomonas aeruginosa* has heterogeneous or homogeneous resistance to polymyxins is in lines 149 - 154.

The MICs of colistin and tigecycline (TGC) for all five CRECC strains were determined using the broth microdilution method, revealing high resistance to colistin (MIC >128 mg/L) and susceptibility to TGC. These strains exhibited homogeneous resistance to colistin (Table 1), as no "skip-well" phenomenon was observed during testing. However, this method may

not detect extremely low-frequency resistant subpopulations, potentially overlooking heteroresistance(42).

Reference:

[42] Alousi S, Saad J, Panossian B, Makhlof R, Khoury CA, Rahy K, Thoumi S, Araj GF, Khnayzer R, Tokajian S. 2024. Genetic and structural basis of colistin resistance in *Klebsiella pneumoniae*: Unravelling the molecular mechanisms. *J Glob Antimicrob Resist* 38:256-264.

4. Lines 252-263: How do you explain the fact that the CRECC402R strain has become sensitive to beta-lactams? Has it lost its plasmid conferring resistance?

Thank you for pointing out these problems. Since our focus is on investigating the reasons for changes in colistin sensitivity following tigecycline resistance in bacteria, it is important to note that bacterial evolution is inherently unstable. Phenotypic changes in response to other classes of drugs may not be constant and can involve evolutionary trade-offs. In our study, the CRECC402R strain regained sensitivity to β -lactam antibiotics, which may be associated with the loss of plasmids carrying β -lactamase genes.

Supplemented in lines 198 - 203 :

Notably, The MICs of meropenem and doripenem for CRECC402R decreased significantly from initial values of ≥ 16 mg/L and ≥ 8 mg/L to ≤ 0.125 mg/L (≥ 128 -fold reduction) and ≤ 0.125 mg/L (≥ 64 -fold reduction), respectively. Additionally, CRECC402R exhibited increased susceptibility to piperacillin, ticarcillin/clavulanic acid, and third-generation cephalosporins (e.g., cefotetan and cefoxime)(Table 1). These changes may be associated with the loss of plasmids carrying β -lactamase genes(42).

Reference:

[42] Alousi S, Saad J, Panossian B, Makhlof R, Khoury CA, Rahy K, Thoumi S, Araj GF, Khnayzer R, Tokajian S. 2024. Genetic and structural basis of colistin resistance in *Klebsiella pneumoniae*: Unravelling the molecular mechanisms. *J Glob Antimicrob Resist* 38:256-264.

5. Lines 267-271: no mutations were found in any of the genes mentioned (table 2), with the exception of ramR or ramA?

Thank you for your careful reminder. We acknowledge the limitations in this area, but we ensured the rigor of our experimental process. The *ramA* gene acts as a bridge in the changes in sensitivity to tigecycline and colistin. Its mutation significantly upregulates the expression

of both itself and the efflux pump genes it regulates, thereby causing tigecycline resistance. Additionally, RamA is known to directly bind and activate genes involved in lipid A biosynthesis, such as *lpxC*, *lpxL-2*, and *lpxO*. This binding induces modifications in the lipid A moiety of lipopolysaccharide, leading to reduced susceptibility to colistin. This mechanism has been described in the literature and is specifically addressed in lines 386-423 of our manuscript. This highlights the complex regulatory network associated with RamA. Similarly, mutations in the *ramR* gene can also explain these related phenomena.

Supplement the text: Lines 211-212:

However, no mutations were found in any of the genes mentioned, with the exception of *ramR* or *ramA*.

6. Line 280: the fold change observed is not quantified in the text. it would be advisable to revise figure 3 and use *Log10* as the ordinate?

Thank you for your insightful suggestion. In the figure, asterisks (*) denote the significance of changes to facilitate clear comparison. Specific fold changes are reported in the text for genes with substantial alterations. For genes with inconsistent expression changes or fold changes, annotations are omitted to maintain clarity and conciseness. The y-axis labeling was verified through extensive review of relevant literature and is confirmed to be accurate.

Supplement the changes in multiples at lines 306, 310, 311 and 356.

7. Line 320: *E. cloacae* reference missing (doi: 10.1128/AAC.00962-20)

We sincerely appreciate the valuable comment. The revised content of the paper is as follows, Cited this literature(line267):

RamA can directly regulate multidrug resistance efflux pumps AcrAB and OqxAB in *E. bugandensis*(43).

Li B, Zhang J, Li X. 2022. A comprehensive description of the TolC effect on the antimicrobial susceptibility profile in *Enterobacter bugandensis*. *Front Cell Infect Microbiol* 12:1036933.

8. Lines 320-327: This has already been noted in the following publication: doi: 10.1128/AAC.00962-20

Thank you for your reminder. The reference has been changed to this one, which is located on line 355.

9. *Lines 327-329: fold changes are not indicated?*

Thank you for pointing out the problem.

The revised sentence is in lines 274 - 275:

Additionally, we observed changes in the expression of *phoQ* (0.1- to 0.9-fold) and *arnA* (0.1- to 0.5-fold) , with almost all TGC-resistant strains exhibiting lower levels compared to parental strains.

Minor comments:

10. - *Line 65 and throughout the text: please write "in vitro" in italics*

Thanks for your careful checks. "vitro" has been italicized in red on line 35.

11. - *Line 68 and throughout the text: please write "Enterobacter" in italics*

Thank you for your careful reminder. "Enterobacter" has been italicized in red on line 38.

12. - *Line 101 and throughout the text: please replace "Klebsiella pneumoniae" by "K. pneumoniae"*

Thanks for your careful checks. The revised content is highlighted in red on line 73.

13. - *Line 242 and throughout the text: please replace "metal-beta lactamase gene" by "metallo-beta-lactamase gene"*

Thanks for your careful checks. The revised content is highlighted in red on line 163.

14. - *the abbreviations COL and TGC are not used throughout the text! please correct!*

Thanks for your careful checks. The abbreviations COL and TGC have been consistently used throughout the text in red. Located at lines 194, 207, 226, 271, 285, 288, 291, 302, 334.

Closing comment to Reviewer 1: We hope the revised manuscripts is now acceptable to you. If not, we are glad to receive any further feedback which we shall continue to apply our best effort to address.

Response to reviewer #2 Comments:

“General comments: In their article entitled "Study on collateral sensitivity of tigecycline to colistin resistant Enterobacter cloacae complex" Kaixin Yu and colleagues explored the mechanism behind the sensitivity to colistin associated with the emergence of tigecycline resistance among five carbapenem resistant Enterobacter cloacae complex clinical isolates. The article is interesting, and explore how the development of resistance to one molecule can transform an initially resistant strain to sensitive considering another molecule. The author explored the duo "colistin - tigecycline". In general, English could be improved as some sentences were difficult to understand. Scientifically speaking, I had some questions/remarks which are detailed below.”

Response: The reviewers were highly professional and must have spent a considerable amount of time reviewing my manuscript. After revising the paper according to their suggestions, the quality of the manuscript has significantly improved. This not only benefits my current research but also plays a crucial role in deepening my understanding of this field for future studies. We appreciate your valuable comment. We have reviewed the entire manuscript and revised any unclear sentences to enhance clarity. Thank you very much.

Major comments:

1. Line 244: *if the mcr variant was the mcr-9 which is not described as responsible for high colistin MICs, do you have an explanation about the original colistin MICs among the clinical isolates?*

We sincerely thank the reviewer for careful reading. As suggested by the reviewer, we have corrected the texts.

The reasons for the resistance of these strains to colistin have been supplemented in lines 166 - 180 .

In our previous study, the *mcr-9* gene in CRECC402 and CRECC405 strains was located on the IncHI2/2A plasmid, with a genetic context of *IS903B-mcr-9-wbuC-IS26*. In CRECC414, the *mcr-9* gene was found on the pECL414-1 plasmid, flanked by *IS1B-mcr-9.2-wbuC-IS26(11)*. In CRECC416, the *mcr-9* gene was located upstream of *IS903B* and downstream of *IS481* and *IS26(44)*. These insertion sequences may facilitate the stable expression and

dissemination of the *mcr-9* gene, potentially serving as key factors in the spread of colistin resistance among CRECC strains in hospital settings. Additionally, the high-level colistin resistance observed in CRECC401 and CRECC402 clinical isolates may be attributed to other mechanisms, such as the overexpression of efflux pumps (e.g., AcrAB-TolC) and biofilm formation. *Enterobacter cloacae* can expel colistin from cells via efflux pump systems, reducing its intracellular accumulation and conferring resistance (45). Overexpression of regulatory genes such as *soxS* and *sdmA* can further enhance resistance by activating efflux pump systems (46).

References:

- [11] Jiang S, Wang X, Yu H, Zhang J, Wang J, Li J, Li X, Hu K, Gong X, Gou X, Yang Y, Li C, Zhang X. 2022. Molecular antibiotic resistance mechanisms and co-transmission of the *mcr-9* and metallo- β -lactamase genes in carbapenem-resistant *Enterobacter cloacae* complex. *Front Microbiol* 13:1032833.
- [44] Li W, Zhang J, Fu Y, Wang J, Liu L, Long W, Yu K, Li X, Wei C, Liang X, Wang J, Li C, Zhang X. 2024. In vitro and in vivo activity of ceftazidime/avibactam and aztreonam alone or in combination against *mcr-9*, serine- and metallo- β -lactamases-co-producing carbapenem-resistant *Enterobacter cloacae* complex. *European Journal of Clinical Microbiology & Infectious Diseases* 43:1309-1318.
- [45] Mousavi SMJ, Hosseinpour M, Rafiei F, Mahmoudi M, Shahraki H, Shiri H, Hashemi A, Sharahi JY. 2025. Colistin antibacterial activity, clinical effectiveness, and mechanisms of intrinsic and acquired resistance. *Microb Pathog* doi:10.1016/j.micpath.2025.107317:107317.
- [46] Telke AA, Olaitan AO, Morand S, Rolain JM. 2017. *soxRS* induces colistin hetero-resistance in *Enterobacter asburiae* and *Enterobacter cloacae* by regulating the *acrAB-tolC* efflux pump. *J Antimicrob Chemother* 72:2715-2721.

2. Line 255: was the decrease of the colistin MICs associated to a loss of plasmid carrying the *mcr* gene?

Thank you very much for the suggestion put forward by the experts. Is the decrease in the MIC value of polymyxin related to the loss of plasmids carrying the *mcr* gene. Plasmid elimination experiments showed that the loss of the *mcr* gene reduced the MIC value of bacteria to colistin from >256 mg/L to 4 mg/L. The added content is in lines 191 - 194.

The reduction in COL MIC values may be due to the loss of *mcr* gene-carrying plasmids, as evidenced by plasmid curing experiments showing a decrease from >256 mg/L to 4 mg/L (47, 48). COL may also kill TGC-resistant mutants by binding to lipid A and disrupting their outer membrane integrity (49).

Reference:

- [47] Liu JH, Liu YY, Shen YB, Yang J, Walsh TR, Wang Y, Shen J. 2024. Plasmid-mediated colistin-resistance genes: *mcr*. Trends Microbiol 32:365-378.
- [48] Li Y, Dai X, Zeng J, Gao Y, Zhang Z, Zhang L. 2020. Characterization of the global distribution and diversified plasmid reservoirs of the colistin resistance gene *mcr-9*. Sci Rep 10:8113.
- [49] Sabnis A, Hagart KL, Klöckner A, Becce M, Evans LE, Furniss RCD, Mavridou DA, Murphy R, Stevens MM, Davies JC, Larrouy-Maumus GJ, Clarke TB, Edwards AM. 2021. Colistin kills bacteria by targeting lipopolysaccharide in the cytoplasmic membrane. Elife 10.

3. Lines 261 - 262: *"The accessory sensitivity induced by TGC exposure was prominently accessory sensitivity induced by TGC exposure was prominently reflected in colistin, while changes in resistance to other antibiotic classes were inconsistent"* when the MICs of quinolones did not change with the tigecycline resistance could it be explain by mutations within QRDR? Did you looked at these mutations?

Thank you for raising this professional question. When the MIC values of quinolones remain unchanged in the context of tigecycline resistance, this may be due to the absence of mutations in the QRDR region. Mutations in the QRDR are the primary mechanism of quinolone resistance, while the overexpression of efflux pumps has a lesser impact on quinolone resistance. Therefore, mutations in the QRDR can explain why the MIC values of quinolones remain unchanged despite tigecycline resistance. This proposal is valuable, and we may subsequently employ gene knockout or overexpression techniques to investigate the impact of specific genes on antibiotic resistance. Determine the cause by sequencing and analyzing the QRDR region or detecting the expression level of efflux pumps.

Reference: 10.1186/s12879-019-4606-y

4. Line 286 [...] but was downregulated in CRECC414R, correlating with a missense mutation in *ramA* (Fig 3). How could you explain the up regulation of *acrB* if *RamA* was absent considering that no other regulator has been found upregulated?

Thank you for raising this question. It is indeed worthy of discussion. Lines 314-319:

In CRECC414R strains, *acrA* and *acrB* genes are upregulated, but *ramA* expression may not align with them. This suggests that *acrAB-tolC* could be regulated independently, possibly by factors like *soxRS*, which can significantly upregulate *acrA*, *acrB*, and *tolC* without significantly affecting *ramA* levels(46). One previous study found that an *E. cloacae* isolate overexpressed *acrA* despite reduced *ramA* levels, with unchanged tigecycline MICs and

overexpression of *rara* and *oqxA*(60).

Reference:

[46] Telke AA, Olaitan AO, Morand S, Rolain JM. 2017. *soxRS* induces colistin hetero-resistance in *Enterobacter asburiae* and *Enterobacter cloacae* by regulating the *acrAB-tolC* efflux pump. *J Antimicrob Chemother* 72:2715-2721.

[60] Pérez A, Poza M, Aranda J, Latasa C, Medrano FJ, Tomás M, Romero A, Lasa I, Bou G. 2012. Effect of Transcriptional Activators SoxS, RobA, and RamA on Expression of Multidrug Efflux Pump AcrAB-TolC in *Enterobacter cloacae*. *Antimicrobial Agents and Chemotherapy* 56:6256-6266.

5. Figure 4 is very difficult to read and understand due to the five strains were not all visible in every time points. The general idea is really good but need to be improve.

Thank you for your reminder. In order to facilitate reading and understanding, the horizontal coordinate uses rectangles to represent generation 0 and generation 200, instead of just using a single coordinate point for display, so as to avoid the overlap of graphics. The modified graph is as follows.

Lines378-381:

Fig 4. Stability test of tigecycline-resistant mutant strains. (A) Changes in colistin MICs; (B) Changes in tigecycline MICs. The X-axis shows the number of generations. The horizontal coordinate uses rectangles to represent generation 0 and generation 200.

6. Figure 3: what is your explanation about the transcriptomic response of *phoQ* and *arnA* associated with tigecycline resistance? What is the promoter of these genes? If they are known, it could be very interesting to study their transcriptomic response associated with tigecycline resistance.

Thank you for your insightful comment and kind suggestion, your question is very insightful.

Supplement the explanations. Except, our study has only preliminarily explored the molecular mechanisms underlying the changes in colistin sensitivity induced by tigecycline. Moving forward, if the promoter sequences of the *phoQ* and *arnA* genes are known, we plan to compare the transcriptomic differences between resistant and susceptible strains under tigecycline stress. This will be our next step. We can then introduce mutations in the promoter regions to observe their effects on gene expression and antibiotic resistance.

7. *Could you please discuss the fact that transcriptomic response of phoQ and arnA are not homogenous among the clinical strains i.e for some of them CRECC401, CRECC402 the gene arnA is down regulated among tigecycline resistant strains while in CRECC405 arnA is up regulated despite the same phenotype: decrease resistance to colistin.*

Thank you for your suggestions. It is necessary to discuss this issue. The relevant explanations are in lines 284 - 294 of the discussion.

The colistin resistance pattern of the *ECC* is closely associated with the presence of the *arnbcadtef* gene cassette (*arn*). The expression of *arn* is regulated by the two-component systems *PmrAB* and *PhoPQ*, as well as the *PhoPQ* feedback inhibitor *MgrB*. However, there may be additional unknown factors or regulatory pathways influencing the expression of *phoQ* and *arnA*, leading to inconsistencies in their expression levels among clinical strains(55). For example, in some strains (e.g., *CRECC401R* and *CRECC402R*), *arnA* is downregulated in isolates, while in others (e.g., *CRECC405R*), *arnA* is upregulated, despite sharing the same phenotype of decreased colistin resistance. In the *CRECC402R* strain, *phoQ* expression remained largely unchanged, whereas *arnA* expression was upregulated in the *CRECC405R* strain relative to the original strain.

Reference: Doijad SP, Gisch N, Frantz R, Kumbhar BV, Falgenhauer J, Imirzalioglu C, Falgenhauer L, Mischnik A, Rupp J, Behnke M, Buhl M, Eisenbeis S, Gastmeier P, Götz H, Häcker GA, Käding N, Kern WV, Kola A, Kramme E, Peter S, Rohde AM, Seifert H, Tacconelli E, Vehreschild M, Walker SV, Zweigner J, Schwudke D, Chakraborty T. 2023. Resolving colistin resistance and heteroresistance in *Enterobacter* species. *Nat Commun* 14:140.

8. *Lines 327-329: "Additionally, we observed changes in the expression of phoQ and arnA, with almost all TGC-resistant strains exhibiting lower levels compared to parental strains." Almost all represented 3/5 strains, what about the two others?*

Thank you for your reminder.

In the CRECC402R strain, *phoQ* expression remained largely unchanged, whereas *arnA* expression was upregulated in the CRECC405R strain relative to the original strain.

9 Regarding the CRECC416, how could you explain the decreased sensitivity of colistin while there was no transcriptomic modification in *phoQ* and *arnA*?

Thank you for your insightful question.

In the CRECC416R strain, the expression levels of *phoQ* and *arnA* did not decrease significantly, yet colistin sensitivity was restored. This could be because in an environment without colistin or other antibiotic selection pressures, the plasmid carrying the *mcr-9* gene is likely to be lost, as maintaining the plasmid may impose a fitness cost on the bacteria(56). Additionally, the modification of the lipid A portion of LPS can mediate RamA expression and lead to an increase in colistin sensitivity(57).

Reference:

[56] Guo Z, Feng S, Liang L, Wu Z, Min L, Wang R, Li J, Zhong LL, Zhao H, Chen X, Tian GB, Yang JR. 2024. Assessment of the reversibility of resistance in the absence of antibiotics and its relationship with the resistance gene's fitness cost: a genetic study with *mcr-1*. *Lancet Microbe* 5:100846.

[57] Yu W, Jia P, Chu X, Li S, Jia X, Zhu Y, Liu X, Xu Y, Yang Q. 2024. Dual role of *ramR* mutation in enhancing immune activation and elevating eravacycline resistance in *Klebsiella pneumoniae*. *iMetaOmics* 1.

10. Line 299 "*Colistin resistance patterns shifted from sensitive to resistant or remained unchanged in these strains (Figure 4).*" But the MICs of colistin were lower than the original strains MICs > 128 mg/L? could you please comment?

Sorry, this sentence is unclear. The revised sentence is in lines 244 - 248.

CRECC401R and CRECC416R mutants shifted from induced susceptibility to COL resistance without restoring initial MIC levels, potentially due to evolutionary trade-offs and fitness costs after acquiring TGC resistance, despite no significant growth rate differences(51). Other mutants remained susceptible (Figure 4).

[51] Andersson DI, Hughes D. 2010. Antibiotic resistance and its cost: is it possible to reverse resistance? *Nat Rev Microbiol* 8:260-71.

11. Times kill curves: Could you please add the number 3 in the y abscises in order to see the bactericidal effect of the combination

Thank you for reminding. We have refined the original figure by adding tick marks on the Y-axis and vertical dashed lines perpendicular to the Y-axis, along with the inclusion of the number "3," as shown in the figure below. The updated image has been replaced in both the clean manuscript and the version displaying tracked changes.

Fig 5. Time killing curve of 5 CRECC strains. The combination of TGC and colistin could be bactericidal. COL, colistin; TGC, tigecycline.

12. Lines 351 - 354 "Similarly, in CRECC414 and 352 CRECC414R, frameshift mutations occur due to the insertion of a single base between 353 base pairs 126 and 127 of the ramR gene, and this region of the ramA gene sequence 354 may be a popular mutation site" What are the consequences in the RamR protein of the frame shift mutations?

Thank you for your question. After determination, the functional domain of the ramR protein has not changed.

13. Line 363 - 368 "CRECC402R and 364 CRECC405R did not have any mutations in the ramR gene, and their ramA gene expression levels were still significantly increased, so there may be other relevant regulatory factors. Several studies have similarly reported *E. cloacae* isolates showing elevated levels of ramA expression without any associated changes in ramR, suggesting the presence of multiple ramA regulatory pathways(43)"

Gravey F et al. in Central role of the ramAR locus in the multidrug resistance in

*ESBL-Enterobacterales. Microbiol Spectr*12:e03548-23 and Baucheron S et al in *Binding of the RamR Repressor to Wild-Type and Mutated Promoters of the ramA Gene Involved in Efflux-Mediated Multidrug Resistance in Salmonella enterica Serovar Typhimurium. Antimicrob Agents Chemother*56: have shown the importance of the intergenic region between *ramR* and *ramA*, could you look at any genomic mutations?

Thank you very much for the constructive feedback. I apologize for not checking the genomic mutations in that region. This content has been added to lines 336 to 342 in the discussion section.

Research had indicated the intergenic region (IR) between *ramR* and *ramA* is vital for regulating *ramA* expression and bacterial multidrug - resistance. Specific IR sequences are crucial for RamR binding; mutations (e.g., substitutions, deletions) disrupt this binding, causing *ramA* overexpression and efflux-pump activation, enhancing resistance. This is prominent in clinically - isolated multidrug-resistant *K. pneumoniae*, *E. cloacae*, and *Salmonella*(61). Moreover, *ramR* mutations reduce its affinity for the *ramA* promoter(57).

Reference:

[61] Gravey F, Michel A, Langlois B, Gérard M, Galopin S, Gakuba C, Du Cheyron D, Fazilleau L, Brossier D, Guérin F, Giard JC, Le Hello S. 2024. Central role of the ramAR locus in the multidrug resistance in ESBL-Enterobacterales. *Microbiol Spectr* doi:10.1128/spectrum.03548-23:e0354823.

[57] Yu W, Jia P, Chu X, Li S, Jia X, Zhu Y, Liu X, Xu Y, Yang Q. 2024. Dual role of *ramR* mutation in enhancing immune activation and elevating eravacycline resistance in *Klebsiella pneumoniae*. *iMetaOmics* 1.

14. Lines 369 - 371 "Previous studies have shown that RamA can activate *lpxC*, *lpxL*, and *lpxO* associated with lipid A biosynthesis, while lipid A modification can lead to a variety of outcomes, such as reduced colistin sensitivity (44-46)." In this publication De Majumdar et al *Elucidation of the RamA regulon in Klebsiella pneumoniae reveals a role in LPS regulation. PLoS pathogens*, 11(1), e1004627. The increased of RamA is associated with increase of colistin MICs, could you please comment?

Thank you for your insightful question. RamA can also reduce colistin sensitivity by modulating the modification of lipid A. This study focuses on the accessory collateral sensitivity of bacteria to colistin and does not involve validation of the above research findings.

Lines 343-344: Previous studies have shown that *ramA* can directly bind to and activate the expression of the promoters of the *lpxC*, *lpxL-2* and *lpxO* genes.

Minor comments:

15. Lines 62 - 63: please add references about « collateral sensitivity»

Thank you for your reminder. Listed at line 63.

References:

[1] Mahmud HA, Wakeman CA. 2024. Navigating collateral sensitivity: insights into the mechanisms and applications of antibiotic resistance trade-offs. *Front Microbiol* 15:1478789.

16. Lines 71 - 74: *there are other new options against CRECC like aztreonam-avibactam or cefiderocol, could you please discuss these molecules?*

Of course, the newly added content is in lines 44 - 48.

K. pneumoniae can develop resistance to CZA through multiple mechanisms, primarily involving mutations in KPC-2 and OmpK36(5). Similarly, mutations in the PiuC, PiuA, and PirA genes of *P. aeruginosa* carrying both IMP-16 and KPC-2 carbapenemase genes can confer in vivo resistance to cefiderocol(6).

Reference:

[5] Xiang X, Kong J, Zhang J, Zhang X, Qian C, Zhou T, Sun Y. 2025. Multiple mechanisms mediate aztreonam-avibactam resistance in *Klebsiella pneumoniae*: Driven by KPC-2 and OmpK36 mutations. *Int J Antimicrob Agents* 65:107425.

[6] Viñes J, Herrera S, Vergara A, Roca I, Vila J, Aiello TF, Martínez JA, Del Río A, Lopera C, Garcia-Vidal C, Casals-Pascual C, Soriano À, Pitart C. 2025. Novel PiuC, PirA, and PiuA mutations leading to in vivo cefiderocol resistance progression in IMP-16- and KPC-2-producing *Pseudomonas aeruginosa* from a leukemic patient. *Microbiol Spectr* 13:e0192824.

17. Line 80: *Are tetracycline and colistin really widespread in Human medicine? Line 82: could you please precise the variant of mcr which is associated to CRECC?*

Thank you for pointing out these problems. Upon verification, it was found not to be widely used. The field of "in humans and animals" was removed, and the mcr variants related to CRECC were clearly identified in the revised manuscript and clean text at line 53.

The widespread use of this cationic antimicrobial peptide has led to the emergence of colistin resistance globally.

18. Lines 104 - 105 *"CRECC poses greater complexity compared to carbapenem--resistant Klebsiella pneumoniae, further research in this area remains limited." Could you please explain why ?*

This statement lacks better literature support, and I apologize for this scientific negligence,

the sentence has been deleted from the manuscript.

19. Lines 163 - 164: *"All strains before and after induction were detected by polymerase chain reaction (PCR) and sanger sequencing." I don't understand this sentence*

Thanks for your careful checks. This sentence is not rigorous enough and has been revised to
“All strains before and after induction were tested for gene mutation by Sanger sequencing.”
(Line98)

20. Lines 197 - 200: *could you please explain the relationship between stability of MICs and growth curves?*

Thanks for your careful checks. This sentence is intended to convey that in order to evaluate the changes in growth adaptability of TGC-resistant mutants compared with their parental strains, the modified content is located at Lines 116 - 118:

“To evaluate the growth fitness alteration of TGC-resistant mutants in comparison to their parental strains,”

21. Lines 213: *in the Time Kill Curve part, could you please explain the dilution ration of 1:100?*

Thank you for your reminder. The revised sentence is located on line 130.

“the dilution ratio of bacterial cultures to broth is 1:100.”

22. Line 235: *what was the method used to assess the genomic distance between the five strains? Could you please provide the Hoffman hsp60 cluster to your five strains?*

The phylogenetic tree was constructed using the neighbor-joining method in MEGA software. Since I have left the laboratory, the data of bacterial genomes are not saved with me, and their phylogenetic relationships are not the focus of the article. If you think it is not rigorous enough, this part can be removed.

23. Line 239: *These are not the epidemic sequence types of ECC infection in China or our hospital. Could you please cite the species and the MLST of the epidemic strains?*

Thank you for your reminder. The additional content is located on lines 160-161 of the

manuscript.

These are not the epidemic sequence types ST114, ST93, ST90, and ST78 associated with ECC infections(43).

Reference:

Mavroidi A, Froukala E, Tsakris A. 2024. Comparative Genomics of an Emerging Multidrug-Resistant bla(NDM)-Carrying ST182 Lineage in Enterobacter cloacae Complex. Antibiotics (Basel) 13.

24. Line 298 *"However, CRECC401R and CRECC416R maintained their resistance to tigecycline" due to fixed ramR mutations, this should be mentioned.*

Thank you for your reminder. This sentence has been added to lines 243.

“This resistance is attributed to a stable mutation in the *ramR* gene. “

25. Line 317 - 319 *"Consequently, uncovering and demonstrating the collateral sensitivity of TGC in colistin-resistant CRECC holds immense potential for novel clinical interventions." Could you please be more prudent about this sentence and rephrase? Thank you.*

Thank you for your suggestion. To ensure greater prudence, the revised sentence is as follows.

Lines 264-366: Therefore, identifying and confirming the collateral sensitivity of TGC in colistin-resistant CRECC could provide promise for the development of new clinical strategies.

26. Line 320 -321: *"RamA can directly regulate multidrug resistance efflux pumps AcrAB and OqxAB in 321 K. pneumoniae(39) » The effect of RamA on AcrAB have also been published in Ecc strains, could you please add appropriate references.*

Thank you for your kind suggestion.

Lines 267: RamA can directly regulate multidrug resistance efflux pumps AcrAB and OqxAB in *E. bugandensis*(53).

Reference:

Li B, Zhang J, Li X. 2022. A comprehensive description of the TolC effect on the antimicrobial susceptibility profile in Enterobacter bugandensis. Front Cell Infect Microbiol 12:1036933.

Closing comment to Reviewer 2: We tried our best to improve the manuscript and made some changes marked in red in revised paper which will not influence the content and framework of the paper. We appreciate for Reviewers' warm work earnestly, and hope the

correction will meet with approval. Once for your comments and again, thank you very much suggestions.

We sincerely thank the reviewers for their voluntary and meticulous reviews, as well as for the valuable comments they have provided. These comments are thought-provoking and have deepened our understanding of this study. The questions raised by you have provided additional directions for our future research and have helped us to consider issues more comprehensively. We are also grateful for the hard work of the editor and for the opportunity to revise our manuscript.

We tried our best to improve the manuscript and made some changes marked in red in revised paper which will not influence the content and framework of the Paper. And we sent the manuscript to experts for language polishing, and the revised manuscript is better than before! We appreciate for Reviewers' warm work earnestly, and hope the correction will meet with approval. This ensures that the first author will meet the requirements for my upcoming master's degree graduation. Best regards!

Sincerely,

Zhang Xiaoli

Re: Spectrum03310-24R1 (**Study on collateral sensitivity of tigecycline to colistin resistant *Enterobacter cloacae* complex**)

Dear Dr. Xiaoli Zhang:

Thank you for the privilege of reviewing your work. Below you will find my comments, instructions from the Spectrum editorial office, and the reviewer comments.

Revision Guidelines

Sincerely,
Siu-Kei Chow
Editor
Microbiology Spectrum

Reviewer #1 (Comments for the Author):

General comments:
Answers to questions are satisfactory and relevant

Minor comments:

- Line 76 and throughout the text: please write genes in italics: "in the *PiuC*, *PiuA*, and *PirA* genes"
- Line 78 and throughout the text: please write "in vivo" in italics
- Line 268 and throughout the text: please write "in vitro" in italics
- Line 366 and throughout the text: please replace "arnbcadtef gene cassette" by "the *arnBCADTEF* operon »

Response Letter

Dear editors and reviewers,

Manuscript number: Spectrum03310-24-R1

Title: Study on collateral sensitivity of tigecycline to colistin resistant *Enterobacter cloacae* complex

We appreciate the opportunity to revise our manuscript titled "Study on collateral sensitivity of tigecycline to colistin resistant *Enterobacter cloacae* complex" and are grateful for the insightful comments provided by the reviewers. In the following, we have provided detailed responses to reviewer's comments. Revised portions are marked in red in the paper. In this response letter, the reviewers' comments are presented in italicized font and special concerns have been numbered. Our corresponding changes and additions to the manuscript are highlighted in red text in "Marked-Up_Manuscript-without_figures2". The response is given in normal font and changes. We have tried our best to make all the revisions clear, and we hope that the revised manuscript meets the requirements for publication.

Response to reviewer #1 Comments

General comments:

Answers to questions are satisfactory and relevant

Minor comments:

Marked-Up_Manuscript-without_figures2

- Line 76 and throughout the text: please write genes in italics: "in the *PiuC*, *PiuA*, and *PirA* genes"

Thank you for your careful reminder. "*PiuC*, *PiuA*, and *PirA* genes" has been italicized in red on lines 76-77.

- Line 78 and throughout the text: please write "in vivo" in italics

Thanks for your careful checks. "vivo" has been italicized in red on line 78.

- Line 268 and throughout the text: please write "*in vitro*" in italics

Thanks for your careful checks. The revised content is highlighted in red on line 268.

- Line 366 and throughout the text: please replace "*arnbcadtef gene cassette*" by "*the arnBCADTEF operon* »

Thanks for your careful checks. The revised content is highlighted in red on line 366.

Closing comment to Reviewer 1: We hope the revised manuscripts is now acceptable to you.

We sincerely thank the reviewers for their voluntary and meticulous reviews, as well as for the valuable comments they have provided. These comments are thought-provoking and have deepened our understanding of this study. We are also grateful for the hard work of the editor and for the opportunity to revise our manuscript.

We tried our best to improve the manuscript and made some changes marked in red in revised paper which will not influence the content and framework of the Paper. We appreciate for Reviewers' warm work earnestly, and hope the correction will meet with approval. This ensures that the first author will meet the requirements for my upcoming master's degree graduation. Best regards !

Sincerely,

Zhang Xiaoli

Re: Spectrum03310-24R2 (**Study on collateral sensitivity of tigecycline to colistin resistant *Enterobacter cloacae* complex**)

Dear Dr. Xiaoli Zhang:

Thank you for the privilege of reviewing your work. Below you will find my comments, instructions from the Spectrum editorial office, and the reviewer comments.

The format of genes is still incorrect. Genes in lines 76-77 should be *piuC piuA pirA* in italics. Please double check all necessary formats of genes, special characters, species, etc.

Revision Guidelines

Sincerely,
Siu-Kei Chow
Editor
Microbiology Spectrum

Response Letter

Dear editors and reviewers,

Manuscript number: Spectrum03310-24-R3

Title: Study on collateral sensitivity of tigecycline to colistin resistant *Enterobacter cloacae* complex

We appreciate the opportunity to revise our manuscript titled "Study on collateral sensitivity of tigecycline to colistin resistant *Enterobacter cloacae* complex" and are grateful for the insightful comments provided by the reviewers. In the following, we have provided detailed responses to reviewer's comments. Revised portions are marked in red in the paper. In this response letter, the reviewers' comments are presented in italicized font and special concerns have been numbered. Our corresponding changes and additions to the manuscript are highlighted in red text in "Marked-Up_Manuscript-without_figures2". The text in the "Manuscript_Text_File" has also been updated to reflect the changes. The italics have been checked, as well as the accuracy of the bacterial species and gene names. The response is given in normal font and changes. We have tried our best to make all the revisions clear, and we hope that the revised manuscript meets the requirements for publication.

Response to Comments

*The format of genes is still incorrect. Genes in lines 76-77 should be *piuC piuA pirA* in italics. Please double check all necessary formats of genes, special characters, species, etc.*

Thank you for your comments. All necessary genes, special characters, species, and other formats that have been checked are listed below.

The font and format have been revised after self-checking.

1. "*PiuC, PiuA, and PirA genes*" has been italicized in red on lines 76-77. It has been changed to "the *PiuC, PiuA* and *PirA* genes"
2. Line 62: "Collateral sensitivity" is changed to "Collateral sensitivity (CS)" on its first occurrence, and all subsequent occurrences of "Collateral sensitivity" are replaced with "CS." Additionally, "collateral sensitivity (CS)" is corrected to "CS." in line 107.

3. Lines 68-69: The phrase "often termed" can be changed to "often referred to as," which is more in line with academic writing standards.

4. Change 'ECC' in line 128 to the same format as 'ECC' in line 70, removing the italics.

5. Line 74: "*K. pneumoniae*" has been changed to "*Klebsiella pneumoniae (K. pneumoniae)*," as this is the first occurrence of this bacterial species in this context.

6. Line 132: "vitro" has been corrected to italic.

7. Line 249: The format of "*IS903B-mcr-9-wbuC-IS26*" has been corrected as "*IS903B-mcr-9-wbuC-IS26*".

8. Line 251: The format of "*IS1B-mcr-9.2-wbuC-IS26*" has been corrected as "*IS903B-mcr-9.2-wbuC-IS26*"

Line 252: Corrected to "*IS903B*", "*IS481*", "*IS26*"

9. Line 258: "*Enterobacter cloacae*" has been formatted consistently with the previous text as "*E. cloacae*."

10. Line 426: "*A. baumannii*" is corrected to "*Acinetobacter baumannii*".

We sincerely thank the reviewers for their voluntary and meticulous reviews, as well as for the valuable comments they have provided. These comments are thought-provoking and have deepened our understanding of this study. We are also grateful for the hard work of the editor and for the opportunity to revise our manuscript.

We tried our best to improve the manuscript and made some changes marked in red in revised paper which will not influence the content and framework of the Paper. We appreciate for Reviewers' warm work earnestly, and hope the correction will meet with approval. This ensures that the first author will meet the requirements for my upcoming master's degree

graduation. Best regards !

Sincerely,

Zhang Xiaoli

Re: Spectrum03310-24R3 (**Study on collateral sensitivity of tigecycline to colistin resistant *Enterobacter cloacae* complex**)

Dear Dr. Xiaoli Zhang:

Your manuscript has been accepted, and I am forwarding it to the ASM production staff for publication. Your paper will first be checked to make sure all elements meet the technical requirements. ASM staff will contact you if anything needs to be revised before copyediting and production can begin. Otherwise, you will be notified when your proofs are ready to be viewed.

Sincerely,
Siu-Kei Chow
Editor
Microbiology Spectrum